# On the scaling of wind turbine rotors

Helena Canet[1], Pietro Bortolotti[2], and Carlo L. Bottasso[1]

[1]Wind Energy Institute, Technische Universität München, 85748 Garching b. München, Germany
[2]National Renewable Energy Laboratory, Golden, CO 80401, USA

**Correspondence:** Carlo L. Bottasso (carlo.bottasso@tum.de)

**Abstract.**

This paper formulates laws for scaling wind turbine rotors. Although the analysis is general, the article primarily focuses on the subscaling problem, i.e. on the design of a smaller size model that mimics a full-scale machine. The present study considers both the steady-state and transient response cases, including the effects of aerodynamic, elastic, inertial and gravitational forces. The analysis reveals the changes to physical characteristics induced by a generic change of scale, indicates which characteristics can be matched faithfully by a subscaled model, and states the conditions that must be fulfilled for desired matchings to hold.

Based on the scaling laws formulated here, the article continues by considering the problem of designing scaled rotors that match desired indicators of a full-scale reference. To better illustrate the challenges implicit in scaling and the necessary trade-offs and approximations, two different approaches are contrasted. The first consists in a straightforward geometric zooming. An analysis of the consequences of zooming reveals that, although apparently simple, this method is often not applicable in practice, because of physical and manufacturing limitations. This motivates the formulation of scaling as a constrained optimal aerodynamic and structural matching problem of wide applicability.

Practical illustrations are given considering the scaling of a large reference 10 MW wind turbine of about 180 m of diameter down to three different sizes of 54, 27 and 2.8 m. Results indicate that, with the proper choices, even models characterized by very significant scaling factors can accurately match several key performance indicators. Additionally, when an exact match is not possible, relevant trends can at least be captured.

## 1 Introduction

This article is concerned with the aeroservoelastic scaling of wind turbine rotors. The general scaling problem includes both up- and sub- (or down-) scaling. This work primarily focuses on the latter aspect, i.e. on the design of subscaled models, but briefly touches also upon the former. Specifically, this work tries to answer the following scientific questions:

- What are the effects of a change of scale (i.e. both in the case of up- and subscaling) on the steady and transient response of a wind turbine?

- What steady and transient characteristics of the response of a full-scale wind turbine can be matched by a subscaled model?

- What are the most suitable ways to design the aerodynamic and structural characteristics of a subscaled model?

The understanding of both up- and subscaling is relevant to contemporary wind energy technology.

Regarding upscaling, wind turbines have experienced a continuous growth in size in the past decades. This trend has been mostly driven by a desire for increased capacity factors, which can be obtained essentially through two main design parameters: by lowering the specific power, which —for a given power rating— means a larger rotor swept area, and by designing taller towers, which reach higher above ground where wind blows faster. In turn, improved capacity factors have a positive effect on the cost of energy, which has helped propel the penetration of wind in the energy mix. The design of the next-generation wind turbines, especially for offshore applications, is expected to follow this same path, with rotor diameters of present and future products already exceeding 200 m (IRENA, 2019; GE, 2019; Siemens Gamesa, 2020). Unfortunately, larger blades cannot be obtained by simply scaling-up existing smaller blades, but must be designed to beat the cubic law of growth. In fact, weight (and hence cost) grows with volume, i.e. with the cube of size, whereas power capture only grows with rotor swept area, i.e. with the square of size (Sieros et al., 2012). Within this background, it is clearly useful to understand the changes that can be expected in a turbine response as the result of an increase in size.

Subscaling, on the other hand, is useful as a research tool: by designing and testing smaller-scale versions of full-scale references, one can validate simulation tools, explore ideas, compare alternative solutions and deepen the knowledge and understanding of complex physical phenomena. Among other advantages, scaled testing is usually much cheaper and less risky than full-scale testing. In addition, full-scale testing is typically performed on prototypes or even commercial products, which raises often unsurmountable issues because of intellectual property rights and trade secrecy. In turn, this limits opportunities for publication, data sharing and for a full exploitation of the results from the scientific community. With commercial turbine sizes expected to grow even further in the future, it becomes more important than ever to fully understand how to best employ subscaling as a research tool.

Two subscaled testing activities are possible: wind tunnel testing with small-scale models, and field testing with small turbines. In both cases, the goal is to match at least some of the characteristics of the original full-scale problem. Clearly, this requires a complete understanding of the effects of a change (in this case, a reduction) of scale on the response of a wind turbine.

Wind tunnel testing of subscaled wind turbine models offers some unique opportunities. First, the operating conditions in a wind tunnel are to a large extent controllable, and typically highly repeatable. Second, measurements –especially of flow quantities— that are possible in the lab environment are generally more difficult, less precise and with a lower resolution at full scale. Third, costs and risks are much more limited than in the case of field testing, and the time for the conduction of the experiments is shorter (because of the reduced challenges, but also because of time acceleration, as explained later). Fourth, since a small-scale model cannot exactly match a full-scale product, property right issues are typically much less of a constraint.

The first wind tunnel experiments on wind turbine aerodynamics were conducted in the last decades of the 20th century, as summarized in Vermeer et al. (2003). Studies carried out during the Unsteady Aerodynamics Experiment (Simms et al., 2001) with a 10 m-diameter, stall-regulated 20 kW turbine were, among others, key to uncovering the importance of specific flow phenomena, such as dynamic stall, 3D rotational effects and tower-wake interactions. Later, the 4.5-m-diameter scaled models designed for the Model rotor EXperiments In controlled COnditions (MEXICO) project enabled the validation of multiple

aerodynamic models, ranging from blade element momentum (BEM) to computational fluid dynamics (CFD) (Snel et al., 2009). These wind turbine models were designed following a set of scaling laws aimed at replicating as accurately as possible the aerodynamic behavior of full-scale machines. More recently, the inclusion of closed loop controls and aeroservoelastic considerations in the scaling process expanded the scope of wind tunnel testing beyond aerodynamics (Campagnolo et al., 2014). Nowadays, wind tunnel tests are extensively used to gain a better understanding of wake effects, to validate simulation tools and to help develop novel control strategies (Bottasso and Campagnolo, 2020). The recent study of Wang et al. (2020) tries to quantify the level of realism of wakes generated by small-scale models tested in a boundary layer wind tunnel.

Unfortunately, the exact matching of all relevant physical processes between full-scale and subscale models is typically not possible. This mismatch increases with the scale ratio and it becomes especially problematic when large wind turbines (with rotor sizes in the order of $10^2$ meters, and power ratings in the order of $10^6$–$10^7$ W) are scaled to very small-size wind tunnel models (characterized by rotors in the order of $10^{-1}$–$10^0$ meters, and power ratings in the order of $10^0$–$10^2$ W). To limit the scale factor, instead of using very small models in a wind tunnel, testing can be conducted in the field with small-size wind turbines (with a rotor in the order of $10^1$ m, power ratings in the order of $10^5$ W).

Examples of state-of-the-art experimental test sites realized with small-size wind turbines are the Scaled Wind Farm Technology (SWiFT) facility in Lubbock, Texas (Berg et al., 2014), which uses three 27-meter diameter Vestas V27 225 kW turbines, or the soon-to-be-ready Winsent complex-terrain facility in the German Swabian Alps (ZSW, 2016), which uses two 54-meter diameter S&G 750 kW turbines.

Reducing the scaling ratios and moving to the field offers the opportunity to overcome some of the constraints typically present in wind tunnel testing, although some of the advantages of wind tunnels are clearly lost. Indeed, the range of testing conditions cannot be controlled at will, measurements are more difficult, and costs are higher. Here research has so far mainly focused on steady-state aerodynamics and wake metrics. For example, within the National Rotor Testbed project (Resor and Maniaci, 2013), teams at the University of Virginia, Sandia National Laboratories and the National Renewable Energy Laboratory have designed a blade for the SWiFT experimental facility, replacing the original Vestas V27 blade. The scaling laws were specifically chosen to replicate the wake of a commercial 1.5 MW rotor at the subscale size of the V27 turbine. To capture the dynamic behavior of very large wind turbines, additional effects must however be considered in the scaling laws. For example, Loth et al. (2017) have recently proposed a methodology to include gravity in the scaling process, and they have demonstrated their approach to scale a 100 m blade down to a 25 m size. Gravity is also crucially important in floating offshore applications (Azcona et al., 2016) to balance buoyancy and correctly represent flotation dynamics, with its effects on loads, stability and performance and with implications in control design.

This paper considers the general problem of scaling a wind turbine rotor to a different size, including the effects caused by aerodynamic, elastic, inertial and gravitational forces. The study is structured in two main parts.

Initially, an analysis of the problem of scaling is presented. The main steady and transient characteristics of a rotor in terms of performance, aeroservoelasticity and wake shedding are considered, and the effects caused by a generic change of scale are determined. The analysis reveals that, in principle, most of the response features can be faithfully represented by a subscaled model. However, an exact matching of all features is typically impossible because of chord-based Reynolds effects, which

lead to changes in the aerodynamic behavior of the system. Another limit comes from wind conditions: the wind field is not scaled when using utility-size models in the field, and wind tunnel flows can only partially match the characteristics of the atmospheric boundary layer. The analysis also shows that scaling is essentially governed by two parameters: the geometric (length) scaling factor and the time scaling factor. Based on these two parameters, all matched and unmatched quantities can be fully characterized.

In a second part, the paper continues by looking at the problem of designing a subscaled model. Two different approaches are considered. The first is a straightforward zooming-down of all blade characteristics based on a pure geometrical scaling (Loth et al., 2017), which is appealing for its apparent simplicity. The second is based on a complete aerostructural redesign, which is formulated here in terms of two constrained optimizations: the aerodynamic one defines the external shape of the blade, whereas the structural optimization sizes the structural components. Both strategies aim at replicating the dynamic behavior (including gravitational effects) of a full-scale wind turbine at a smaller scale, and they are therefore based on the same scaling laws. Clearly, the complete redesign is a more complicated process than the pure geometric zooming-down approach. However, the main goal of scaling is that of designing a rotor that matches at scale the behaviour of a target full-scale machine as closely as possible. From this point of view, the simplicity of design —which is a one-off activity— is less of a concern, especially today, when sophisticated automated rotor design tools are available (Bortolotti et al., 2016). Apart from simplicity, zooming is very often simply not possible for large scale factors because of unrealistically small sizes (especially the thickness of shell structures), non-achievable material characteristics, or impossible to duplicate manufacturing processes (Wan and Cesnik, 2014; Ricciardi et al., 2016). In all those cases, a different aerodynamic shape, structural configuration and materials are used to obtain the desired behavior, as shown, for example, in the design of a small-size aeroelastically-scaled rotor by Bottasso et al. (2014), or as customarily done in the design of scaled flutter models for aeronautical applications (Busan, 1998).

Although the intrinsic limits of the straightforward zooming down approach are probably well understood, these two alternative methodologies are compared here in order to give a better appreciation of the complexities that one has to face in the design of scaled models. To give practical and concrete examples, a very large rotor is scaled down to three different model sizes, including two different utility wind turbines and a small-scale wind tunnel model. For each model, the zooming down approach is adopted when possible for its simplicity, and then replaced by the re-design method when fidelity or physical limits make it impractical or impossible.

Furthermore, the paper analyses the accuracy with which the subscale models successfully mirror relevant key characteristics of the full-scale reference, both in terms of absolute values and of trends. This is indeed an important aspect of scaling: even if the exact matching of certain quantities is sometimes not possible, scaled models can still be highly valuable if they are able to at least capture trends. As an example of such a trend analysis, the subscale models are used here to explore changes in loading between unwaked and waked inflow conditions, which are then validated against the corresponding loading changes of the full-scale machine. Results indicate that even the smallest model is capable of capturing complex details of wake interaction, including an interesting lack of symmetry for left/right wake impingements caused by rotor uptilt.

A final section completes the paper, listing the main conclusions that can be drawn from the results and highlighting their limits.

## 2    Scaling

Buckingham's Π Theorem (Buckingham, 1914) states that a scaled model (labelled $(\cdot)_M$) has the same behavior as a full-scale physical system (labelled $(\cdot)_P$) if all the $m$ relevant nondimensional variables $\pi_i$ are matched between the two systems. In

other words, when the governing equations are written as

$$\phi(\pi_{1P}, \ldots, \pi_{mP}) = 0, \tag{1a}$$

$$\phi(\pi_{1M}, \ldots, \pi_{mM}) = 0, \tag{1b}$$

then the two systems are similar if

$$\pi_{iP} = \pi_{iM}, \quad i = (1, m). \tag{2}$$

Depending on the scaled testing conditions, not all dimensional quantities can usually be matched. In the present case, we consider that testing is performed in air, either in a wind tunnel or in the field, neglecting hydrodynamics.

The length (geometric) scale factor between scaled and full-scale systems is defined as

$$n_l = \frac{l_M}{l_P}, \tag{3}$$

where $l$ is a characteristic length (for example the rotor radius $R$), whereas the scale factor for time $t$ is defined as

$$n_t = \frac{t_M}{t_P}. \tag{4}$$

As a consequence of these two definitions, one can determine the angular velocity and wind speed scaling factors, which respectively write $n_\Omega = \Omega_M/\Omega_P = 1/n_t$ and $n_v = V_M/V_P = n_l/n_t$. A nondimensional time can be defined as $\tau = t\Omega_r$, where $\Omega_r$ is a reference rotor speed; for example, the rated one. It is readily verified that, by the previous expressions, nondimensional time is matched between the model and physical system, i.e. $\tau_M = \tau_P$. The two factors $n_l$ and $n_t$ condition, to a large extent,

the characteristics of a scaled model.

The following Sects. 2.1 and 2.2 analyze the main steady and transient characteristics of a rotor in terms of performance, aeroservoelasticity and wake shedding. The analysis discusses which of these characteristics can be matched by a scaled model, and which conditions are required for the matchings to hold. Next, Sect. 2.3 offers an overview on the main scaling relationships and discusses the choice of scaling parameters.

### 2.1    Steady state

#### 2.1.1    Rotor aerodynamics

The power coefficient characterizes the steady-state performance of a rotor, and it is defined as $C_P = P/(1/2\rho A V^3)$, where $P$ is the aerodynamic power, $\rho$ the density of air, $A = \pi R^2$ the rotor disk area and $V$ the ambient wind speed. The thrust

coefficient characterizes the wake deficit and the rotor loading and is defined as $C_T = T/(1/2\rho A V^2)$, where $T$ is the thrust
force. For a given rotor, the power and thrust coefficients depend on tip-speed ratio (TSR) $\lambda = \Omega R/V$, and blade pitch $\beta$, i.e.
$C_P = C_P(\lambda, \beta)$ and $C_T = C_T(\lambda, \beta)$.

It is readily verified that $\lambda_M = \lambda_P$ for any $n_l$ and $n_t$, which means that it is always possible to match the scaled and full-scale
TSR. This ensures the same velocity triangle at the blade sections and the same wake helix pitch.

Ideally, a scaled model should match the $C_P$ and $C_T$ coefficients of a given full-scale target; it is clearly desirable for the
match not to hold at a single operating point, but over a range of conditions. BEM theory (Manwell et al., 2002) shows that
both rotor coefficients depend on the steady-state aerodynamic characteristics of the airfoils. In turn, the lift $C_L$ and drag $C_D$
coefficients of the aerodynamic profiles depend on the angle of attack, and on the Mach and Reynolds numbers.

The local Mach number accounts for compressibility effects, and is defined as $\mathrm{Ma} = W/a_s$, where $W$ is the flow speed
relative to a blade section, and $a_s$ is the speed of sound. Using the previous expressions, the Mach number of the scaled model
is $\mathrm{Ma}_M = \mathrm{Ma}_P\, n_l/n_t^2$. Because of typical tip speeds, compressibility does not play a significant role in wind turbines. Hence,
the matching of the Mach number can be usually neglected for current wind turbines. The situation might change for future
offshore applications where, without the constraints imposed by noise emissions, higher tip-speed and TSR rotors may have
interesting advantages.

The Reynolds number represents the ratio of inertial to viscous forces, and is defined as $\mathrm{Re} = \rho l u/\mu$, where $l$ is a charac-
teristic length, $u$ a characteristic speed and $\mu$ the dynamic viscosity. In the present context, the most relevant definition of the
Reynolds number is the one referred to the blade sections, where $l = c$ is the chord length, and $u = W$ is the flow speed relative
to the blade section. In fact, the Reynolds number has a strong effect on the characteristics and behavior of the boundary layer
that develops over the blade surface, which in turn, through the airfoil polars, affects the performance and loading of the rotor.
Testing in air in a wind tunnel or in the field (hence with similar $\rho$ and $\mu$, but with a reduced chord $c$) leads to a mismatch
between the scaled and full-scale chord-based Reynolds numbers, as $\mathrm{Re}_M = \mathrm{Re}_P\, n_l^2/n_t$.

The effects due to a chord-based Reynolds mismatch can be mitigated by replacing the airfoils of the full-scale system with
others better suited for the typical Reynolds conditions of the scaled model (Bottasso et al., 2014). A second approach is to
increase the chord of the scaled model. This, however, has the effect of increasing the rotor solidity —defined as $\Sigma = BA_b/A$,
where $B$ is the number of blades and $A_b$ the blade planform area— which may have additional consequences. In fact, the TSR
of the maximum power coefficient is directly related to rotor solidity. This can be shown by using classical BEM theory with
wake swirl, which gives the optimal blade design conditions by maximizing power at a given design TSR $\lambda_d$. By neglecting
drag, the optimal design problem can be solved analytically to give the chord distribution of the optimal blade along the
spanwise coordinate $r$ (Manwell et al., 2002):

$$\frac{c(r)}{R} = \frac{16\pi}{9 B\, C_L \lambda_d^2\, r/R}. \tag{5}$$

Although based on a simplified model that neglects some effects, this expression shows that chord distribution and design
TSR are linked. This means that, if one increases solidity (and hence chord) to contrast the Reynolds mismatch while keeping
$C_L$ fixed, the resulting rotor will have a lower TSR for maximum power coefficient. Therefore, this technique of correcting

the Reynolds moves the optimal TSR away from the one of the full-scale reference, which may or may not be acceptable, depending on the goals of the model. For example, if one wants to match the behavior of the $C_P - \lambda$ curves over a range of TSRs, such an approach would not be suitable. As shown by Eq. (5), this effect can be eliminated or mitigated by changing the design $C_L$ accordingly; however, if this moves the operating condition of the airfoil away from its point of maximum efficiency, a lower maximum power coefficient will be obtained.

In addition, chord $c$ and lift $C_L$ are further constrained by the circulation $\Gamma = 1/2 c C_L W$ (Burton et al., 2001), which plays an important role in the aerodynamics of the rotor and its wake.

Considering first the rotor, the lift and drag generated by the airfoils located close to the blade root are modified by the combined effects of centrifugal and Coriolis forces. In fact, the former cause a radial pumping of the flow that, as a result, moves outboard in the spanwise direction. This radial motion over a rotating body generates chordwise Coriolis forces that alleviate the adverse pressure gradient on the airfoils and, in turn, delay stall. As shown by the dimensional analysis developed by Dowler and Schmitz (2015), rotational augmentation causes multiplicative corrections, noted $g_{C_L}$ and $g_{C_D}$, to the nonrotating lift and drag coefficients that can be written, respectively, as

$$g_{C_L} = \left(\frac{c}{r}\right)^2 \left(\frac{\Gamma}{RW}\right)^{1/2} \left(\frac{\Omega r}{2W}\right)^{-2}, \tag{6a}$$

$$g_{C_D} = \frac{1}{3} \left(\frac{r}{R}\right) \left(\frac{c}{r}\right)^{-1} \left(\frac{\mathrm{d}\theta}{\mathrm{d}r} \frac{R}{\Delta\theta}\right) \left(\frac{\Omega r}{2W}\right), \tag{6b}$$

where $\Delta\theta$ is the total blade twist from root to tip. Equations (6) show that, in order to match the effects of rotational augmentation, the model and full-scale system should have the same blade nondimensional chord and twist distributions, the same nondimensional circulation $\Gamma/(RW)$, and the same Rossby number $\mathrm{Ro} = \Omega r/(2W)$, which represents the ratio of inertia to Coriolis forces. Matching nondimensional circulation between the two systems implies either matching both the planform shape $c/R$ and the lift coefficient $C_L$, or the product of the two. As previously noted, some of these options may lead to a different TSR of optimal $C_P$. On the other hand, it is readily verified that the Rossby number is always matched for any choice of $n_l$ and $n_t$.

### 2.1.2 Wake aerodynamics

The circulation is not only relevant for rotational augmentation but also for wake behavior. In fact, each blade sheds trailing vorticity that is proportional to the spanwise gradient $\mathrm{d}\Gamma/\mathrm{d}r$ (Schmitz, 2020). Therefore, designing a blade that matches the spanwise distribution of $\Gamma$ (and, hence, also its spanwise gradient) ensures that the scaled rotor sheds the same trailed vorticity. Additionally, a matched circulation ensures also a matched thrust, which is largely responsible for the speed deficit in the wake and for its deflection in misaligned conditions (Jiménez et al., 2010).

The Reynolds mismatch derived earlier applies also to its rotor-based definition, which is relevant to wake behavior and is obtained by using $l = 2R$ and $u = V$. However, Chamorro et al. (2012) showed that the wake is largely unaffected by this parameter as long as $\mathrm{Re} > 10^5$, which is typically the case unless extremely small model turbines are used. The same is true for

the terrain-height-based Reynolds definition that applies to flows over complex terrains, where Reynolds-independent results are obtained when $\mathrm{Re} > 10^4$ (McAuliffe and Larose, 2012).

The detailed characterization of the behavior of scaled wakes is considered as out of the scope for the present investigation, and the interested reader is referred to Wang et al. (2020) for a specific study on this important topic.

### 2.1.3 Gravity

The Froude number represents the ratio of aerodynamic to gravitational forces and writes $\mathrm{Fr} = V^2/gR$, where $g$ is the acceleration of gravity. The Froude number of the scaled model is readily found to be $\mathrm{Fr}_M = \mathrm{Fr}_P n_l/n_t^2$. Enforcing Froude ($\mathrm{Fr}_M = \mathrm{Fr}_P$) results in the time scaling factor being set to $n_t = \sqrt{n_l}$. This condition determines the only remaining unknown in the scaling laws, so that the scalings of all nondimensional parameters can now be expressed in terms of the sole geometric scaling factor $n_l$. Froude scaling is used when gravity plays an important role, for example in the loading of very large rotors or for floating offshore applications where weight and buoyancy forces should be in equilibrium.

### 2.1.4 Elasticity

The steady deflections due to aerodynamic loading of the scaled and full-scale wind turbines can be matched by adjusting the stiffness of the scaled model. In fact, consider the very simplified model of a blade represented by a clamped beam of length $R$ under a uniform distributed aerodynamic load per unit span, noted $q = 1/2\,\rho W^2 cC_L$. The beam nondimensional tip deflection is $s/R = qR^3/(8\,EJ)$, where $EJ$ is the bending stiffness, $E$ is Young's modulus and $J$ is the cross-sectional moment of inertia. By the previous definitions of length and time scales, one gets that $(s/R)_M = (s/R)_P$ if $(EJ)_M = (EJ)_P\, n_l^6/n_t^2$. Hence, nondimensional deflections can be matched, provided that the stiffness can be adjusted as shown. Matching this requirement may imply changing the material and/or the configuration of the structure, because of technological, manufacturing and material property constraints (Busan, 1998; Ricciardi et al., 2016), as discussed more in detail later on.

## 2.2 Transient response

A scaled model should obey some additional conditions in order for the transient response of the full-scale system to be matched.

### 2.2.1 Rotor aerodynamics and inflow

As mentioned earlier, any aerodynamically scaled model can always be designed to enforce the TSR without additional conditions. To extend the similitude to dynamics, the nondimensional time derivative of the TSR should also be matched, i.e. $\lambda'_M = \lambda'_P$, where a nondimensional time derivative is noted as $(\cdot)' = \mathrm{d}\cdot/\mathrm{d}\tau$. By using the definition of $\lambda$ one gets

$$\lambda' = \frac{\Omega' R}{V} - \lambda\frac{V'}{V}. \tag{7}$$

The rotor dynamic torque balance equilibrium writes $I\dot{\Omega} = Q$. In this expression, $I$ is the rotor polar moment of inertia, $(\dot{\cdot}) = \mathrm{d}\cdot/\mathrm{d}t$ indicates a derivative with respect to time, and $Q = Q_a - (Q_e + Q_m)$ is the shaft torque. The aerodynamic torque

is noted as $Q_a = 1/2\rho A R C_P/\lambda$, while $Q_e$ is the electrical torque provided by the generator and $Q_m$ the mechanical losses. The aerodynamic torque scales as $Q_{a_M} = Q_{a_P} n_l^5/n_t^2$, and clearly $Q_e + Q_m$ must scale accordingly. Since the mechanical losses depend on friction, it might be difficult to always match $Q_m$, especially in a small-scale model. This problem, however, can be eliminated by simply providing the necessary electrical torque to generate the correct term, $Q_e + Q_m$. By considering that the dimensions of $I$ are $[I] = [\rho_m][l]^5$, where $\rho_m$ is the material density and $l$ a characteristic length, the first term $\Omega'R/V$ in Eq. (7) is matched between the two models if the material density is matched, i.e. if $\rho_{m_M} = \rho_{m_P}$.

The second term $\lambda V'/V$ in Eq. (7) is matched if the two systems operate at the same TSR and if the wind speed has the same spectrum of the wind in the field. The matching of wind fluctuations (clearly, only in a statistical sense) induces the same variations in the TSR, and hence in the rotor response, but also the same recovery of the wake, which is primarily dictated by the ambient turbulence intensity (Vermeer et al., 2003).

Matching of the wind spectrum is in principle possible in a boundary layer wind tunnel, if a flow of the desired characteristics can be generated. Turbulent flows can be obtained by active (Hideharu, 1991; Mydlarski, 2017) or passive means (Armitt and Counihan, 1968; Counihan, 1969). Active solutions are more complex and expensive, but also more flexible and capable of generating a wider range of conditions. When testing in the field, the flow is invariably not scaled. This will have various effects on the scaled model response, which might be beneficial or not depending on the goals of scaled testing. In fact, the acceleration of time ($t_M = t_P n_t$) implies a shift in the wind frequency spectrum. Among other effects, this means that low-probability (extreme) events happen more frequently than at full scale. Similarly, the scaling of speed ($V_M = V_P n_l/n_t$) implies higher amplitudes of turbulent fluctuations and gusts than at full scale.

Magnitude and phase of the aerodynamic response of an airfoil (as for example modelled by Theodorsen's theory (Bisplinghoff and Ashley, 2002)) are governed by the reduced frequency $\kappa = \omega_m c/(2W)$, where $\omega_m$ is the circular frequency of motion. Harmonic changes in angle of attack take place at various frequencies $\omega_{m_j}$, and are caused by the inhomogeneities of the flow (shears, misalignment between rotor axis and wind vector), blade pitching and structural vibrations in bending and twisting. The reduced frequency can be written as $\kappa_j = \widetilde{\omega}_{m_j}\Omega c/(2W)$, where $\widetilde{\omega}_{m_j} = \omega_{m_j}/\Omega$ indicates a nondimensional frequency. This expressions shows that once the nondimensional frequencies, $\widetilde{\omega}_{m_j}$ —due to inflow, pitch and vibrations— are matched, also the corresponding reduced frequencies are matched, as the term $\Omega c/(2W)$ is always automatically preserved between scaled and full-scale systems for any $n_l$ and $n_t$.

Dynamic stall effects depend on reduced frequency $\kappa$, and chord-based Reynolds number. Typical dynamic stall models depend on the lift, drag and moment static characteristics of an airfoil and various time constants that describe its unsteady inviscid and viscous response (Hansen et al., 2004). As previously argued, $\kappa$ can be matched, and all time constants are also automatically matched by the matching of nondimensional time. However, a mismatch of the chord-based Reynolds is typically unavoidable and will imply differences in the dynamic stall behavior of the scaled and full-scale models, which will have to be quantified on a case-by-case basis.

### 2.2.2 Wake aerodynamics

The Strouhal number is associated with vortex shedding, which has relevance in tower and rotor wake behavior; the Strouhal number has also been recently used to describe the enhanced wake recovery obtained by dynamic induction control (Frederik et al., 2019). A rotor-wake relevant definition of this nondimensional parameter is $\text{St} = f2R/V$, where $f$ is a characteristic frequency. Using the previous relationships, it is readily shown that $\text{St}_M = \text{St}_P n_l/(n_t n_v) = 1$, i.e. the Strouhal number is always exactly matched between scaled and full-scale models for any $n_l$ and $n_t$ when TSR is matched.

During transients, spanwise vorticity is shed that is proportional to its temporal gradient. Using BEM theory (Manwell et al., 2002, p. 175), the nondimensional spanwise circulation distribution is computed as

$$\frac{\Gamma}{RW} = \frac{1}{2}\frac{c}{R}C_{L,\alpha}\left(\frac{U_P}{U_T} - \theta\right). \tag{8}$$

In this expression, $C_{L,\alpha}$ is the slope of the lift curve, $\theta$ is the sectional pitch angle, while $U_P$ and $U_T$ are the flow velocity components at the blade section, respectively perpendicular and tangent to the rotor disk plane, such that $W^2 = U_P^2 + U_T^2$. The flow speed component tangential to the rotor disk is $U_T = \Omega r + u_T$, where $u_T$ contains terms due to wake swirl and yaw misalignment. The flow speed component perpendicular to the rotor disk is $U_P = (1-a)V + \dot{d} + u_P$, where $a$ is the axial induction factor, $\dot{d}$ the out-of-plane blade section flapping speed, and $u_P$ the contribution due to yaw misalignment and vertical shear. Neglecting $u_P$ and $u_T$ and using Eq. (8), the nondimensional time rate of change of the circulation becomes

$$\frac{\mathrm{d}}{\mathrm{d}\tau}\left(\frac{\Gamma}{RW}\right) = \frac{1}{2}\frac{c}{R}C_{L,\alpha}\frac{\mathrm{d}}{\mathrm{d}\tau}\left(\frac{1 - a + \dot{d}/V}{\lambda}\left(\frac{R}{r}\right) - \theta\right). \tag{9}$$

For a correct similitude between scaled and full-scale systems, the nondimensional derivatives $\lambda'$, $a'$, $\theta'$ and $(\dot{d}/V)'$ should be matched.

The matching of $\lambda'$ has already been addressed. The term $a'$ accounts for dynamic changes in the induction, which are due to the speed of actuation (of torque and blade pitch) and by the intrinsic dynamics of the wake. The speed of actuation is matched if the actuators of the scaled model are capable of realizing the same rates of change of the full-scale system, i.e. if $\theta'$ is matched. The intrinsic dynamics of the wake are typically modelled by a first-order differential equation (Pitt and Peters, 1981):

$$\dot{\boldsymbol{a}} + \boldsymbol{A}\boldsymbol{a} = \boldsymbol{b}, \tag{10}$$

where $\boldsymbol{a}$ represents inflow states and $\boldsymbol{A}$ a matrix of coefficients proportional to $V/R$. It is readily verified that the matching of nondimensional time results in the matching of $a'$. Finally, the term $(\dot{d}/V)'$ is due to the elastic deformation of the blade, which is addressed next.

### 2.2.3 Elasticity

Considering blade flapping, the Lock number Lo is defined as

$$\text{Lo} = \frac{C_{L,\alpha}\rho c R^4}{I_b}, \tag{11}$$

where $I_b$ the blade flapping inertia. Matching the Lock number ensures the same ratio of aerodynamic to inertial forces. Considering that the flapping inertia is dimensionally proportional to $[\rho_m][l]^5$, where $\rho_m$ is the material density and $l$ a characteristic length, matching the Lock number can be obtained by simply matching the material density of the blade, i.e. $\rho_{mM} = \rho_{mP}$. A similar definition of the Lock number can be developed for the fore-aft motion of the rotor due to the flexibility of the tower, leading to the same conclusion.

The system $i$th nondimensional natural frequency is defined as $\widetilde{\omega}_i = \omega_i/\Omega$, where $\omega_i$ is the $i$th dimensional natural frequency. Matching the lowest $N$ nondimensional frequencies means that the corresponding eigenfrequencies in the scaled and full-scale system have the same relative placement among themselves and with respect to the harmonic excitations at the multiple of the rotor harmonics. In other words, the two systems have the same Campbell diagram (Eggleston and Stoddard, 1987). In addition, by matching nondimensional frequencies, the ratio of elastic to inertial forces is correctly scaled. Considering that the bending natural frequency of a blade is dimensionally proportional to $\sqrt{EJ/\rho_m l^6}$, the matching of nondimensional natural frequencies implies $(EJ)_M = (EJ)_P \, n_l^6/n_t^2$, which is the same result obtained in the steady case for the matching of static deflections under aerodynamic loading. The same conclusions are obtained when considering deformation modes other than bending, so that in general one can write $K_M = K_P \, n_l^6/n_t^2$ where $K$ is a stiffness. Here again, it can be concluded that for each given $n_l$ and $n_t$, one can match the frequencies by adjusting the stiffness of the scaled model.

It should be remarked that this condition only defines the stiffnesses that should be realized in the scaled model, not how these are actually obtained. As noted earlier, it is typically difficult if not impossible to simply zoom down a complex realistic structure, and the model design may require a different configuration and choice of materials (Busan, 1998). An optimization-based approach to the structural matching problem is described later in this work.

It is worth noting that matching both the Lock number and the placement of nondimensional natural frequencies implies that structural deflections caused by aerodynamic loads are correctly scaled. In fact, the Lock number is the ratio of aerodynamic to inertial forces, while $\widetilde{\omega}_i^2$ is proportional to the ratio of elastic to inertial forces. Therefore, if both ratios are preserved, then $\mathrm{Lo}/\widetilde{\omega}_i^2$, being the ratio of aerodynamic to elastic forces, is also preserved. In symbols, this ratio writes

$$\frac{\mathrm{Lo}}{\widetilde{\omega}_i^2} = \frac{qL^3}{EJ}, \tag{12}$$

where the right-hand side is indeed proportional to the nondimensional tip deflection $\widetilde{s} = s/R$ of a clamped beam subjected to a distributed load $q = C_{L,\alpha}\rho c(R\Omega)^2$.

The matching of frequencies is also relevant to the matching of transient vorticity shedding in the wake, as mentioned earlier. In fact, assume that the blade flapping motion can be expressed as the single mode $d = d_0 e^{\omega_f t}$, where $d$ is the flapping displacement and $\omega_f$ the flapping eigenfrequency. Then, the term $(\dot{d}/V)'$ of Eq. (9) becomes

$$\frac{\mathrm{d}}{\mathrm{d}\tau}\left(\frac{\dot{d}}{V}\right) = \frac{d_0}{R}\lambda\tilde{\omega}_f^2 e^{\tilde{\omega}_f \tau}, \tag{13}$$

where $\tilde{\omega}_f = \omega_f/\Omega$ is the nondimensional flapping frequency. This term is matched between the scaled and full-scale models if the nondimensional flapping frequency is matched.

## 2.3 Subscaling criteria

As shown earlier, scaling is essentially governed by two parameters: the geometric scaling factor $n_l$, and the time scaling factor $n_t$. No matter what choice is made for these parameters, the exact matching of some nondimensional parameters can always be guaranteed; these include nondimensional time, TSR, and Strouhal and Rossby numbers. In addition, the matching of other nondimensional quantities can be obtained by properly scaling some model parameters, again independently from the choice of $n_l$ and $n_t$. For example, selecting the material density as $\rho_{mM} = \rho_{mP}$ enforces the matching of the Lock number, whereas scaling the stiffness as $K_M = K_P n_l^6 / n_t^2$ ensures the proper scaling of the system nondimensional natural frequencies. This way, several steady and unsteady characteristics of the full-scale system can be replicated by the scaled system. Other quantities, however, cannot be simultaneously matched, and one has to make a choice.

Table 1 summarizes the main scaling relationships described earlier. The reader is referred to the text for a more comprehensive overview of all relevant scalings.

**Table 1.** Main scaling relationships relevant to a wind turbine. Additional scaling effects are discussed in the text.

| Quantity | Scaling ratio | Coefficient | Comment |
|---|---|---|---|
| Length | $l_M/l_P$ | $n_l$ | |
| Time | $t_M/t_P$ | $n_t$ | |
| Nondimensional time | $\tau_M/\tau_P$ | 1 | |
| TSR $\lambda$ | $\lambda_M/\lambda_P$ | 1 | |
| Rotor speed | $\Omega_M/\Omega_P$ | $1/n_t$ | Due to nondimensional time matching |
| Wind speed | $V_M/V_P$ | $n_l/n_t$ | Due to nondimensional time & TSR matching |
| Mach number | $\mathrm{Ma}_M/\mathrm{Ma}_P$ | $n_l/n_t^2$ | |
| Reynolds number | $\mathrm{Re}_M/\mathrm{Re}_P$ | $n_l^2/n_t$ | |
| Froude number | $\mathrm{Fr}_M/\mathrm{Fr}_P$ | $n_l/n_t^2$ | |
| Strouhal number | $\mathrm{St}_M/\mathrm{St}_P$ | 1 | Due to TSR matching |
| Rossby number | $\mathrm{Ro}_M/\mathrm{Ro}_P$ | 1 | Due to TSR matching |
| Lock number | $\mathrm{Lo}_M/\mathrm{Lo}_P$ | 1 | Requires $\rho_{mM} = \rho_{mP}$ |
| Nondimensional nat. freq. | $\widetilde{\omega}_{iM}^n/\widetilde{\omega}_{iP}^n$ | 1 | Requires $K_M = K_P n_l^6 / n_t^2$ |
| Deflections due to aero. loads | $\widetilde{s}_M/\widetilde{s}_P$ | 1 | Due to Lock & nondim. freq. matching |
| Reduced frequency | $\kappa_{jM}/\kappa_{jP}$ | 1 | Requires $(\widetilde{\omega}_{m_j})_M/(\widetilde{\omega}_{m_j})_P$ due to inflow, pitch and vibrations |
| Nondim. TSR rate of change | $\lambda'_M/\lambda'_P$ | 1 | Requires $(Q_e + Q_m)_M = (Q_e + Q_m)_P n_l^5/n_t^2$, $\rho_{mM} = \rho_{mP}$ and $(V'/V)_M = (V'/V)_P$ |

The choice of the scaling parameters $n_l$ and $n_t$ is highly problem-dependent. Indeed, given a full-scale reference, $n_l$ is set by the size of its scaled replica, which is usually predefined to a large extent. For instance, the choice of the subscale size for a wind tunnel model depends on the characteristics of the target tunnel, to limit blockage (Barlow et al., 1999). When scaling

down to a utility size, one typically chooses to reblade an existing turbine (Berg et al., 2014; Resor and Maniaci, 2013), thereby setting the scaling factor. The choice of $n_t$ is often not straightforward, and typically implies tradeoffs among quantities that cannot all be simultaneously matched.

For example, when the effects of gravity have to be correctly represented by the scaled model, then the matching of the Froude number must be enforced. By setting $\mathrm{Fr}_M = \mathrm{Fr}_P$, one obtains the condition on the time scaling factor $n_t = \sqrt{n_l}$. Having set $n_t$, the scalings of all nondimensional parameters can now be expressed in terms of the sole geometric scaling factor $n_l$.

    Another example is given by the design of small-scale wind turbine models for wind tunnel testing, which typically leads to

small geometric scaling factors $n_l$. Bottasso et al. (2014) defined an optimal scaling by minimizing the error in the Reynolds number and the acceleration of scaled time. The latter criterion was selected to relax the requirements on closed-loop control sampling time: since $\mathrm{Re}_M = \mathrm{Re}_P\, n_l^2/n_t$, small geometric scaling factors might require very fast scaled times and hence high sampling rates, which could be difficult to achieve in practice for closed-loop control models. Bottasso and Campagnolo (2020) used a different criterion, where a best compromise between the Reynolds mismatch and power density is sought. In fact, power

density (defined as power $P$ over volume or, in symbols, $\rho_P = P/R^3$) scales as $\rho_{P_M}/\rho_{P_P} = n_l^2/n_t^3$ and, hence, increases rapidly for small $n_t$. For small $n_l$ it becomes increasingly difficult, if not altogether impossible, to equip the scaled models with functional components (i.e. drive-train, generator, actuation systems, sensors, etc.) that fit in the dimensions prescribed by the scaling factors. The adoption of larger components can be acceptable or not, depending on the nonphysical effects that are generated by their bigger dimensions and the goals of the model.

Yet another example of how delicate these choices can be is found in the experiments described by Kress et al. (2015). In this work, a scaled rotor was designed for experiments in a water tank, with the goal of comparing upwind and downwind turbine configurations. The rotor of the model was scaled geometrically from a full-scale reference; however, the same scaling ratio could not be used for the nacelle because of the need to house the necessary mechanical components. As a result, the model was equipped with an unrealistically large nacelle that, combined with the lower Reynolds number (which causes a thicker

boundary layer), likely increased the redirection of the flow towards the outer blade portions in the downwind configuration. In turn, this led to the conclusion that nacelle blockage improves power production in downwind rotors. Although this may be true for the scaled experiment, there is little evidence that the same conclusion holds for a full-scale machine (Anderson et al., 2020). Because of miniaturization constraints, a larger nacelle is also used in the TUM G1 scaled turbine (Bottasso and Campagnolo, 2020), a machine designed to support wake studies and wind farm control research. The effects of the out-of-scale nacelle on

the wake have however been verified, and appear in this case to be very modest (Wang et al., 2020).

    Additionally, particular combinations of $n_l$ and $n_t$ can make it difficult to find suitable designs. A clear example is found in the structural redesign of an aeroelastically-subscaled blade. Indeed, as previously discussed, the scaled blade should have stiffnesses that scale as $K_M = K_P n_l^6/n_t^2$ and a mass density that scales as $\rho_{mH} = \rho_{mP}$ to ensure the same nondimensional frequencies and Lock number. Depending on the values of the scaling parameters chosen, these scaling relationships might

lead to unconventional combinations of stiffness and mass properties, which can be challenging to fulfill as shown in the next section.

## 3 Design strategies

Upscaling is a design effort driven by different criteria including, among others, annual energy production (AEP), cost of material and manufacturing, logistics and transportation, etc. The situation is different for subscaling. In fact, the previous section has clarified the scaling relationships that exist between a full-scale system and its scaled model. The analysis has revealed that in general several steady and unsteady characteristics of the original system can be preserved in the scaled one. The question is now how to design such a scaled model in order to satisfy the desired matching conditions. This problem is discussed in this section.

### 3.1 Straightforward zooming-down

This approach is based on the exact geometric zooming of the blade, including both its external and internal shape, and it has been advocated by Loth et al. (2017).

Regarding the external blade shape, geometric zooming implies that the same airfoils are used for both the scaled and the full-scale models. The mismatch of the Reynolds number (which is $\mathrm{Re}_M = \mathrm{Re}_P n_l^{3/2}$ for Froude scaling) may imply a different behavior of the polars, especially for large values of $n_l$. On the other hand, as shown earlier, a geometric scaling ensures the near matching (up to the effects due to changes in the polars) of various characteristics, such as optimum TSR, nondimensional circulation, rotational augmentation and vorticity shedding.

Regarding the internal blade shape, the skin, shear webs and spar caps are also geometrically scaled down when using straightfoward zooming. It should be noted that, for large geometric scaling factors $n_l$, the thickness of elements such as the skin or the shear webs may become very thin, possibly less than typical composite plies.

The zoomed scaling has to satisfy two constraints on the properties of the materials used for its realization.

A first constraint is represented by the matching of material density ($\rho_{mM} = \rho_{mP}$), which is necessary to ensure the same Lock number. It should be remarked that the overall material density of the blade includes not only the density of the main structural elements, but also contributions from coatings, adhesive and lightning protection. These components of the blade may not be simply scaled down, so this problem may deserve some attention.

A second constraint is represented by the scaling of the stiffness, which is necessary for ensuring the matching of nondimensional natural frequencies. For Froude scaling, stiffness changes as $K_M = K_P n_l^5$. Considering bending, the stiffness is $K = EJ$. For a blade made of layered composite materials, the bending stiffness is more complicated than the simple expression $EJ$, and it will typically need to be computed with an ad hoc methodology, for example using the anisotropic beam theory of Giavotto et al. (1983). However, the simple expression $EJ$ is sufficient for the dimensional analysis required to understand the effects of scaling. Since the sectional moment of inertia $J$ is dimensionally proportional to $l^4$, $l$ being a characteristic length of the blade cross section, this constraint requires Young's modulus to change according to $E_M = E_P n_l$. This implies that all materials used for the scaled blade, including the core, should have a lower stiffness (and the same density) of the materials used at full scale; as shown later, this constraint is not easily met.

Since strain $\epsilon$ is defined as the ratio of a displacement and a reference length, then it follows that $\epsilon_M = \epsilon_P$. Therefore, given that $E_M = E_P n_l$, then $\sigma_M = \sigma_P n_l$, and the stresses in the scaled model are reduced compared to the ones in the full-scale model. Still, one would have to verify that the admissible stresses and strains of the material chosen for the scaled blade are sufficient to ensure integrity.

The critical buckling stress of a curved rectangular plate is

$$\sigma_{\mathrm{cr}} = k_c \frac{\pi^2 E}{12(1-\nu^2)} \left(\frac{d}{b}\right)^2, \tag{14}$$

where $k_c$ is a coefficient that depends on the aspect ratio of the panel, its curvature and its boundary conditions, $\nu$ is Poisson's ratio, $d$ the panel thickness and $b$ the length of the loaded edges of the plate (Jones, 2006). Here again, the expression of the critical stress of a layered anisotropic composite plate would be more complex than the one reported in Eq. (14), but this is enough for the present dimensional analysis. By using the scaling relationships for length and for $E$, Eq. (14) readily leads to $\sigma_{\mathrm{cr}_M} = \sigma_{\mathrm{cr}_P} n_l$. This means that if the full-scale blade is buckling free, so is the scaled one, as both the critical buckling stress and the stresses themselves scale in the same manner.

## 3.2 Aerostructural redesign

An alternative approach to the design of a subscale model is to identify an external shape and an internal structure that match, as closely as possible, the aeroelastic behavior of the full-scale blade. This approach offers more degrees of freedom, at the cost of an increased design complexity; indeed, one designs a new blade that, although completely different from the full-scale one, matches some of its characteristics.

In this second approach, the first step consists of defining a blade shape that can mimic the aerodynamic behavior of the full-scale system. As previously discussed, this can be obtained according to different criteria. Here, the following three conditions are considered. First, a new set of airfoils is selected to match as closely as possible, despite the different Reynolds of operation, the polar coefficients of the airfoils of the full-scale blade; this is relevant for the matching of the performance and loading of the rotor. Second, the two rotors should have similarly shaped power coefficient curves, which is relevant for performance on and off the design point. Finally, the blades should have the same spanwise circulation distribution, which is relevant for a similar aerodynamic loading of the blade and wake behavior. The resulting scaled blade shape (both in terms of cross sections, because of the changed airfoils, and in terms of chord and twist distributions) will be different from the full-scale rotor. However, this is clearly irrelevant, as the goal is to match some quantities of interest between the two rotors, not their shape.

The aerodynamic design problem can be formally expressed as

$$\min_{\mathbf{p}_a} J_a(\mathbf{p}_a), \tag{15a}$$

$$\text{subject to:} \quad \mathbf{m}_a(\mathbf{p}_a) = 0, \tag{15b}$$

$$\mathbf{c}_a(\mathbf{p}_a) \leq 0. \tag{15c}$$

Vector $\mathbf{p}_a$ indicates the aerodynamic design variables, which include the chord and twist distributions $c(\eta)$ and $\theta(\eta)$, appropriately discretized in the spanwise direction, while $J_a$ is a design figure of merit, $\mathbf{m}_a$ are matching constraints, and finally $\mathbf{c}_a$

are additional design conditions. This formulation of the aerodynamic design problem is very general, and different choices of the figure of merit and of the constraints are possible, depending on the goals of the scaled model.

In the present work, the aerodynamic optimization cost function is formulated as

$$J_a = \sum_i^{N_{C_P}} \left( \frac{C_P(\lambda_i) - \widehat{C}_P(\lambda_i)}{\widehat{C}_P(\lambda_i)} \right)^2 . \tag{16}$$

This cost drives the design towards the power coefficient of the target full-scale model $\widehat{C}_P$ at $N_{C_P}$ control stations. This cost function ensures that the subscale model —whose airfoils generally present a reduced efficiency due to the lower chord-based Reynolds— has a $C_P$ that is as close as possible to the full-scale model. Using $N_{C_P} = 1$, leads to a design with a best $C_P$ at the TSR $\lambda_1$.

Within the vector of matching equality constraints, one set of conditions enforces the matching of the spanwise distribution of the circulation $\widehat{\Gamma}$ at $N_\Gamma$ control stations:

$$\frac{\Gamma(\eta_i) - \widehat{\Gamma}(\eta_i)}{\widehat{\Gamma}(\eta_i)} = 0, \quad i = (1, N_\Gamma), \tag{17}$$

where $\widehat{(\cdot)}$ indicates in general a to-be-matched scaled quantity of the target full-scale model. Another constraint may be added to prescribe the maximum power coefficient to take place at the same design TSR, i.e. $\lambda_{\max(C_P)} = \lambda_{\max(\widehat{C}_P)}$. Finally, vector $\mathbf{c}_a$ specifies additional design inequality constraints, which may include a margin to stall, maximum chord and others, depending on the application.

Once the new aerodynamic shape is identified, the second step consists in the design of an internal blade structure that can mimic the full-scale aeroelastic behavior while ensuring integrity and satisfying manufacturing and realizability constraints. This approach allows for more freedom than the zooming-down approach; for example, one can use different materials than the ones used for the full-scale design, and nonstructural masses can be added without affecting the matching characteristics of the scaled blade.

The structural design problem can be formally expressed as

$$\min_{\mathbf{p}_s} J_s(\mathbf{p}_s), \tag{18a}$$

$$\text{subject to:} \quad \mathbf{m}_s(\mathbf{p}_s) = 0, \tag{18b}$$

$$\mathbf{c}_s(\mathbf{p}_s) \leq 0. \tag{18c}$$

Vector $\mathbf{p}_s$ indicates the structural design variables, which include the size of the various blade structural elements (skin, spar caps, shear webs, leading and trailing edge reinforcements), discretized span- and chordwise. Here again, this formulation is very general, and specific goals will lead to different choices of the merit function and of the constraints.

For example, assuming the blade to be modelled as a beam, the structural optimization cost can be formulated as

$$J_s = \sum_i^{N_s} \left( \frac{M_p(\eta_i) - \widehat{M}_p(\eta_i)}{\widehat{M}_p(\eta_i)} \right)^2 + w_s \sum_i^{N_s} \left( \frac{K_q(\eta_i) - \widehat{K}_q(\eta_i)}{\widehat{K}_q(\eta_i)} \right)^2, \qquad p \in \mathcal{S}_M, q \in \mathcal{S}_K, \tag{19}$$

where $w_s$ is a tuning weight, $M_p$ and $K_q$ are elements of the mass and stiffness matrices, and the sets $\mathcal{S}_M$ and $\mathcal{S}_K$ identify the elements that should be considered within the generally fully populated symmetric mass and stiffness matrices. The first term in the cost aims at the matching of the scaled target mass distribution, while the second at the stiffness distribution. Vector $\mathbf{m}_s$ indicates the matching equality constraints. These may include the matching of a desired number of natural frequencies $\omega_i = \widehat{\omega}_i$, and the matching of a desired number of mode shapes and/or static deflections $\mathbf{u}_j(\eta_i) = \widehat{\mathbf{u}}_j(\eta_i)$ at a given number of spanwise stations $\eta_i$. Finally, vector $\mathbf{c}_s$ specifies the additional design inequality constraints. These constraints express all other necessary and desired conditions that must be satisfied in order for the structural design to be viable, and in general include maximum stresses and strains for integrity, maximum tip deflection for safety, buckling, manufacturing and technological conditions.

It should be noted that the matching of the scaled beam stiffness and mass distributions —if it can be achieved— is an extremely powerful condition. In fact, a geometrically exact non-linear beam model is fully characterized entirely in terms of its reference curve, stiffness and mass matrices (Bottasso and Borri, 1998). This means that exactly matching all of these quantities would ensure the same non-linear structural dynamic behavior of the full-scale target. As shown later, this is not always possible because of limits due to technological processes, material characteristics, chosen configuration of the scaled model, etc. In this case, there is a partial match between the full-scale and scaled beam models, and the sets $\mathcal{S}_M$ and $\mathcal{S}_K$ include only some elements of the mass and stiffness matrices. When this happens, additional matching constraints can help in ensuring as similar a behavior as possible between the scaled and full-scale structures, for example by including static deflections and/or modal shapes, as shown later.

## 4 Application and results: subscaling of a 10 MW rotor

The two strategies of straightforward zooming and aerostructural redesign are applied here to the subscaling of a 10 MW machine, developed in Bottasso et al. (2016) as an evolution of the original Denmark Technical University (DTU) 10 MW reference wind turbine (Bak et al., 2013). The main characteristics of the turbine are reported in Table 2. Some of the principal blade characteristics are given in Table 3, which reports the position of the airfoils, whereas Table 4 details the blade structural configuration and Table 5 summarizes the material properties.

**Table 2.** Principal characteristics of the full-scale 10 MW wind turbine (Bottasso et al., 2016).

| Data | Value | Data | Value |
|------|-------|------|-------|
| Wind class | IEC 1A | Rated electrical power | 10.0 MW |
| Hub height [H] | 119.0 m | Rotor diameter [D] | 178.30 m |
| Cut-in wind speed [$V_{\text{in}}$] | 4 ms$^{-1}$ | Cut-out wind speed [$V_{\text{out}}$] | 25 ms$^{-1}$ |
| Rotor cone angle [$\Xi$] | 4.65 deg | Nacelle uptilt angle [$\Phi$] | 5.0 deg |
| Rotor solidity [$\Sigma$] | 4.66% | Max blade tip speed [$v_{\text{tip}_{\max}}$] | 90.0 ms$^{-1}$ |
| Blade mass | 42,496 kg | Tower mass | 617.5 ton |

**Table 3.** Spanwise position of the airfoils of the blade of the 10 MW machine.

| Airfoil | Thickness | Position | Airfoil | Thickness | Position |
|---------|-----------|----------|---------|-----------|----------|
| Circle | 100.0% | 0.0% | FFA-W3-301 | 30.1% | 38.76% |
| Circle | 100.0% | 1.74% | FFA-W3-241 | 24.1% | 71.87% |
| FFA-W3-480 | 48.0% | 20.80% | FFA-W3-241 | 24.1% | 100.00% |
| FFA-W3-360 | 36.0% | 29.24% | | | |

**Table 4.** Main structural characteristics of the blade of the 10 MW machine.

| Component | From (% span) | To (% span) | Material type |
|-----------|---------------|-------------|---------------|
| External shell | 0 | 100 | Tx GFRP |
| Spar caps | 1 | 99.8 | Ux GFRP |
| Shear web | 5 | 99.8 | Bx GFRP |
| Third shear web | 22 | 95 | Bx GFRP |
| TE/LE reinforcements | 10 | 95 | Ux GFRP |
| Root reinforcement | 10 | 99.8 | Balsa |
| Shell and web core | 5 | 99.8 | Balsa |

**Table 5.** Mechanical properties of the materials of the blade of the 10 MW machine.

| Material type | Longitudinal elasticity modulus [MPa] | Transversal elasticity modulus [MPa] | Density [kgm$^{-3}$] |
|---------------|---------------------------------------|--------------------------------------|----------------------|
| Tx GFRP | 21,790 | 14,670 | 1,845 |
| Ux GFRP | 41,630 | 14,930 | 1,940 |
| Bx GFRP | 13,920 | 13,920 | 1,845 |
| Balsa | 50 | 50 | 110 |

Three different subscalings are considered here. The first subscale model, denominated W-model, is based on the German Winsent test site (ZSW, 2016), which is equipped with two 750 kW turbines with a rotor diameter of 54 m (ZSW, 2017). The reference rotor blades are scaled down to match the span of the Winsent blades; reblading one of the Winsent turbines yields a subscale model of the full-scale 10 MW turbine suitable for field testing. The second model, denominated S-model, is based on the SWiFT test site, which is equipped with Vestas V27 turbines. Here, the full-scale rotor is scaled down to a diameter of 27 m. Finally, the T-model is a wind tunnel model with a rotor diameter of 2.8 m, which is similar to the scaled floating turbine tested in the Nantes wave tank in the INNWIND.EU project (Azcona et al., 2016).

Table 6 reports the different geometric scaling factors and a few additional key quantities of the three subscale models. For all, Froude scaling is used, which sets the timescale factor as previously explained. The application of the scaling laws to the full-scale turbine results in the characteristics listed in Table 7. Independently of the approach chosen to define the internal and external shape, the scaled models must fulfill these conditions to correctly mirror the dynamic behavior of the full-scale wind turbine.

**Table 6.** Some key scaling factors for the W-, S- and T-models.

| Quantity | Scaling factor | W | S | T |
|---|---|---|---|---|
| Length | $n_l$ | 1:3.30 | 1:6.60 | 1:63.68 |
| Time | $\sqrt{n_l}$ | 1:1.82 | 1:2.57 | 1:7.98 |
| Mass | $n_l{}^3$ | 1:36 | 1:288 | 1:258,214 |
| Rotor speed | $\sqrt{n_l}$ | 1:1.82 | 1:2.57 | 1:7.98 |
| Wind speed | $\sqrt{n_l}$ | 1:1.82 | 1:2.57 | 1:7.98 |
| Reynolds | $n_l{}^{3/2}$ | 1:6 | 1:16.97 | 1:508 |
| Stiffness | $n_l{}^5$ | 1:392 | 1:12,558 | 1:32,360 |

**Table 7.** Gravo-aeroservoelastic scaling requirements for the W-, S- and T-models.

| Data | Full scale | W | S | T |
|---|---|---|---|---|
| Diameter [m] | 178.3 | 54.0 | 27.0 | 2.8 |
| Hub height [m] | 119.0 | 36.04 | 18.02 | 1.87 |
| Total blade mass [kg] | 42,496 | 1,180 | 148 | 0.16 |
| Rotor speed [rpm] | 8.9 | 16.2 | 22.9 | 71.1 |
| TSR for max $C_P$ [-] | 7.2 | 7.2 | 7.2 | 7.2 |
| Reynolds [-] | 1E+7 | 1.7E+6 | 5.9E+5 | 2E+4 |
| First flapwise frequency [Hz] | 0.57 | 1.04 | 1.46 | 4.52 |
| First edgewise frequency [Hz] | 0.72 | 1.31 | 1.85 | 5.77 |

The gravo-aeroservoelastic scaling laws lead to very light and flexible subscale blades. For instance, the standard blades of the V27 weigh 600 kg (Vestas, 1994), which is four times more than the gravo-aeroservoelastically scaled blades of the S-model. It should however be remarked that this ratio would be smaller for a modern blade, since the V27 was designed more than 25 years ago and its blades are heavier than the ones based on contemporary technology.

The following sections detail the design of the external and internal shape of the three subscale blades. Section 4.1 describes the aeroservoelastic and design tools used to this end. Then, Sect. 4.2 and 4.3 discuss, respectively, the strengths and limitations of each design strategy for each subscale model.

## 4.1 Aeroservoelastic and design tools

The aeroservoelastic models are implemented in `Cp-Lambda` (Bottasso et al., 2012). The code is based on a multibody formulation for flexible systems with general topologies described in Cartesian coordinates. A complete library of elements, including rigid bodies, nonlinear flexible elements, joints, actuators and aerodynamic models is available, as well as sensor and control elements.

The aerodynamic characteristics of the blade are described through lifting lines, including spanwise chord and twist distribution and aerodynamic coefficients. The code is coupled with aerodynamic models based on the BEM model, formulated according to stream-tube theory with annular and azimuthally-variable axial and swirl inductions, unsteady corrections, root and blade tip losses as well as a dynamic stall model.

The tower and rotor blades are modeled by nonlinear geometrically-exact beams of arbitrary initially undeformed shapes, which are bending, shear, axial and torsion deformable. The structural and inertial characteristics of each beam section are computed with `ANBA` (Giavotto et al., 1983), a 2D finite-element cross-sectional model. Finally, full-field turbulent wind grids are computed with `TurbSim` (Jonkman et al., 2009) and used as input flow conditions for the aeroservoelastic simulations.

`Cp-Max` (Bortolotti et al., 2016) is a design framework wrapped around `Cp-Lambda`, which implements optimization algorithms to perform the coupled aerostructural design optimization of the blades and, optionally, of the tower. For the present work, the code was modified to implement also the scaled design matching optimizations defined by Eqs. (15) and (18). All optimization procedures are solved with a sequential quadratic programming algorithm, in which gradients are computed by means of finite differences.

## 4.2 External shape design

For all three models, the design of the subscale external blade shape aims at replicating the aerodynamic characteristics of the full-scale rotor, including its wake. As long as the chord-based Reynolds numbers are sufficiently large, a zooming-down approach is clearly the simplest strategy for designing the external shape of a scaled blade.

Airfoil FFA-W3-241 equips the outermost part of the full-scale blade (see Table 3). Its performance at the three typical Reynolds numbers of the full-scale, W- and S-models was computed with `ANSYS Fluent` (ANSYS, Inc., 2019). The results are reported in Fig. 1. The performance of the airfoil is clearly affected by the Reynolds number, with a particularly significant drop in efficiency for the lowest Reynolds case. Notwithstanding these Reynolds effects, the zooming-down approach is selected for the W- and S-models, since the airfoils are still performing well at their corresponding typical subscale Reynolds. A redesign approach with alternative airfoils was not attempted here, and would probably lead only to marginal improvements of the aerodynamic performance.

On the other hand, for the small geometric scaling factor of the T-model, the aerodynamic redesign approach is necessary. In general, smooth airfoils present a large reduction in aerodynamic efficiency below a critical Reynolds of about 70,000 (Selig et al., 1995). Efficient profiles specifically developed for low Reynolds applications are generally necessary in order to get a good matching of the full-scale aerodynamic performance. As an alternative to the original airfoil, the 14%-thick

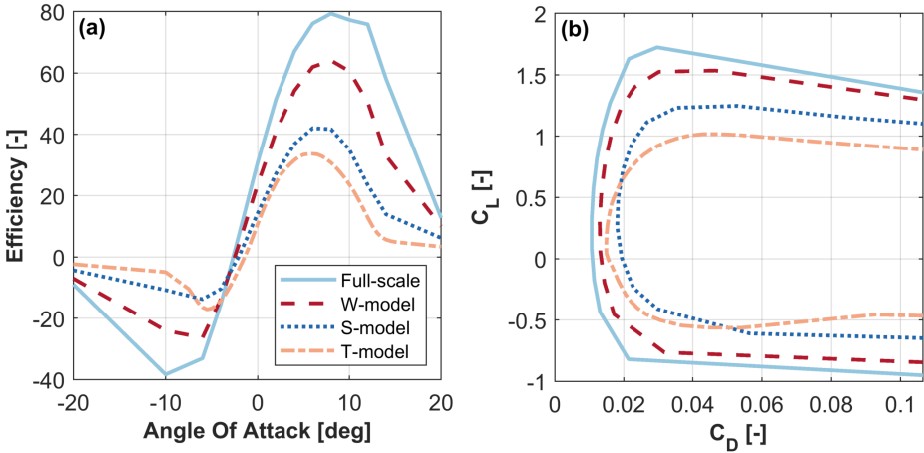

**Figure 1.** Aerodynamic characteristics of the airfoil at the outermost part of the blades at the corresponding Reynolds number. The full-scale, W- and S-models are equipped with the FFA-W3-241 airfoil. The T-model is designed with the RG14 airfoil. (a) Efficiency, $E = C_L/C_D$, vs. angle of attack; (b) polar curves, i.e. $C_L$ vs. $C_D$.

airfoil RG14 (Selig et al., 1995) is selected, because its aerodynamic characteristics at the scaled Reynolds are in reasonable agreement with the ones of the original airfoil at its full-scale Reynolds (Fig. 1). The blade is then completely redesigned, using the RG14 airfoil along its full span.

The blade shape is parameterized by means of chord and twist spanwise distributions. The design problem is formulated as the maximization of the power coefficient at the design TSR $\lambda_d$ of the full-scale rotor, solving Eq. (15) with the cost given by Eq. (16) for $N_{C_P} = 1$ and $\lambda_1 = \lambda_d$. The nonlinear constraints expressed by Eq. (17) enforce the same spanwise nondimensional circulation distribution of the full-scale blade.

Figure 2 shows the external shapes of the full-scale blade and the three subscale models in terms of chord, relative thickness, twist and Reynolds number. Clearly, the shape curves for the W- and S-models overlap with the full-scale ones, because zooming is used in these two cases, as previously explained.

The three subscale models have the same TSR in region II as the full-scale machine, and the correspondingly subscaled rated rotor speeds. The rated wind speeds do not exactly match the subscale ones, on account of the differences in the $C_P$-TSR curves caused by Reynolds effect.

### 4.3 Design of the internal structure

The definition of the internal structure has to achieve the following goals: the matching of the full-scale aeroelastic behavior, the integrity of the blade under loading and the feasibility of the manufacturing process. In the next two sections, the zooming-down and the redesign approaches are applied to the structure of the three subscale blades.

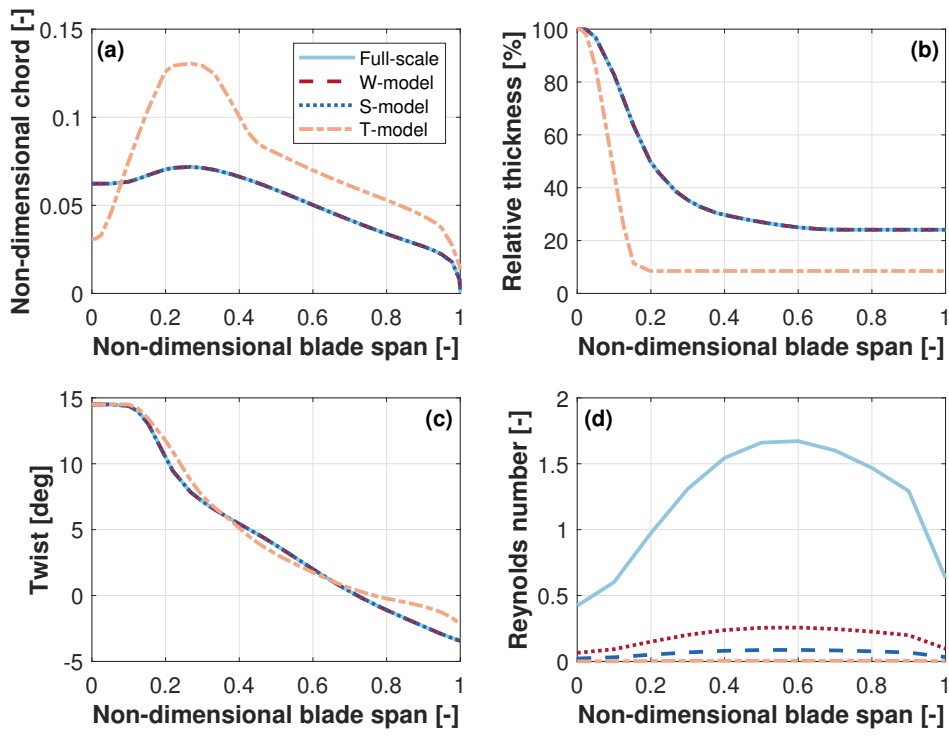

**Figure 2.** (a) Nondimensional chord, (b) relative thickness, (c) twist, and (d) Reynolds number vs. spanwise position, for the full-scale blade and its three subscale models.

### 4.3.1 Limits of the zooming-down approach

The straightforward zooming-down approach can be applied to the internal structure of the W- and S-model blades, as their external geometrical shape has also been defined following this approach. The resulting structures satisfy all scaling constraints, but present some critical challenges.

First, the thicknesses of some of the components are unrealistically low. The blade root of the W-model is, for example, only 20 mm thick and is therefore unable to accommodate the root-bolted connections. Furthermore, the scaling of the outer shell skin leads to a laminate thickness of less than one ply. The third web of the S-model blade is also extremely thin (less than 1 mm) and very close to the trailing edge.

Additionally, the scaled structure requires materials characterized by very peculiar mechanical properties. Indeed, as previously shown, the scaling laws require the modulus of elasticity to obey the relationship $E_M = E_P n_l$, and the material density to be $\rho_{mM} = \rho_{mP}$. For example, the outer shell of the W-model blade requires an elasticity modulus of 6.6 GPa and a density of 1,845 kgm$^{-3}$, which are not typical values of conventional materials (see Fig. 3). Finally, nonstructural masses, such as glue, paint and lightning protection, cannot be exactly zoomed down by geometric scaling, and need to be treated separately.

One may try to relax some of these hurdles by increasing the necessary component thicknesses and choosing materials with mechanical properties that compensate this increase. For example, a threefold increase of the skin thickness in the W-model would be able to accommodate the root-bolted connection and would satisfy manufacturing tolerances. To meet the mass and inertia constraints, a material should be used that has a lower density, $\rho_{mM} = \rho_{mP}/3$, and a lower-elasticity modulus,

$E_M = E_P n_l/3$. Figure 3 reports Ashby's diagram of Young's modulus vs. density (Materials Data Book, 2003). In this plot, the values corresponding to the outer shell skin materials have been marked with $\times$ symbols. A red symbol indicates the full-scale blade, a yellow symbol is used for the W-model considering the exact zooming-down approach, whereas a green symbol indicates the solution with a threefold thickness increase. It should be noted that, although the properties of the scaled models do correspond to existing materials, these are typically not employed for the manufacturing of blades. Therefore, their actual

use for the present application might indeed pose some challenges.

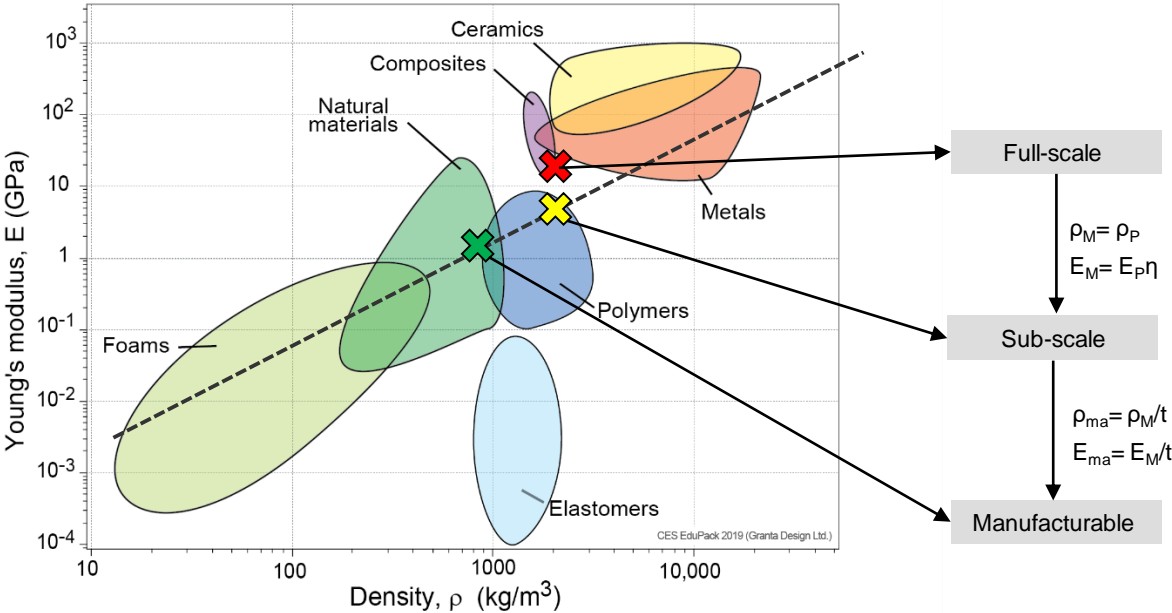

**Figure 3.** Ashby's diagram of Young's modulus vs. density (Materials Data Book, 2003), and the outer shell skin materials for the W-model. Chart created using CES EduPack 2019, ANSYS Granta ©2020 Granta Design.

Overall, the zooming-down approach for the structural design is not really straightforward and is significantly more complicated than in the case of the aerodynamic design. An alternative is offered by a complete redesign of the internal structure, which is illustrated in the next section.

### 4.3.2 Redesign of the W- and S-models

An alternative to the zooming-down approach is the redesign of the internal structure. This consists of a typical blade design process, subjected to additional constraints that enforce the desired scaling relationships but, crucially, subjected also to all other conditions that are necessary to make the design viable. For example, here a lower bound to the thickness of all structural components is set to 1 mm, while a minimum thickness of 60 mm is assumed at the root to accommodate the bolted connection of the W- and S-models.

Additionally, one has a larger freedom in the choice of materials. For the present applications, the glass-fiber-reinforced plastic (GFRP) composites of the full-scale blade appear to be suitable choices also for the W-model. On the other hand, these materials are too stiff for the S-model, due to its smaller geometric scaling. An alternative was found within the family of thermoplastic materials that have typical stiffness values between $1 - 3$ GPa and densities between 900 and 1,400 $\text{kgm}^{-3}$ (Brondsted et al., 2005). Although not strictly of interest here, thermoplastics also have interesting advantages over thermosets,

such as reduced cycle times, lower capital costs of tooling and equipment, smaller energy consumption during manufacturing and enhanced recyclability at the end of their life (Murray et al., 2018).

During the design phase of the subscale models, a more careful attention can also be paid to the distributions of nonstructural masses. Specifically, masses from shell and sandwich cores must be recomputed for the new scaled structure in order to prevent buckling of the sandwich panels. Additional masses from surface finishing and painting are also recomputed according to the

625 surface of the external shell. In fact, if a zooming-down strategy is chosen for the design of the external geometry, these masses will scale with the length scale factor. Masses from resin uptake in the outer shell and shear webs are recomputed for the scaled structure assuming a constant area density. Indeed, this value does not change from the full to the sub scale, since it depends on the material and manufacturing process. A different assumption is taken for the masses of bonding plies and adhesive along the shear webs, leading and trailing edge. Since these masses are chordwise dependent, the linear density of these materials in

the subscale size must be corrected by the length scale factor. Finally, the linear density of the lightning protection system is assumed to be constant for all sizes.

The structural design is formulated as the matching optimization problem expressed by Eq. (18). The cost function given by Eq. (19) considers the sole spanwise matching of the mass distribution, i.e. it neglects inertia terms in $\mathcal{S}_M$ and uses $w_s = 0$. The matching constraints $\mathbf{m}_s$ include the lowest three natural frequencies, and the static deflected shape of the outboard 40%

section of the blade. This static condition was chosen to represent the maximum tip displacement resulting from turbulent simulations in power production for the full-scale machine (design load case DLC 1.1, see IEC (2005)). Finally, the additional design constraints $\mathbf{c}_s$ include stresses, strains, fatigue and technological constraints in the form of bounds on thickness and thickness rate of change of the laminates.

The structural design for the W- and S-models is based on a typical thin-walled composite configuration, where the design

variables are defined as the spanwise thicknesses of the skin, shear webs, spar caps and leading and trailing edge reinforcements. Given the smaller size of the scaled blades, one single shear web is used instead of the three used in the full-scale 10 MW model.

Table 8 describes the mechanical properties of the materials used for these two blades, while Table 9 associates the various structural elements with the materials.

**Table 8.** Mechanical properties of the materials used for the W- and S-model blades.

| Material type | Longitudinal elasticity modulus [MPa] | Transversal elasticity modulus [MPa] | Density [kgm$^{-3}$] |
|---|---|---|---|
| Bx GFRP | 13,920 | 13,920 | 1,845 |
| Ux GFRP | 42,000 | 12,300 | 1,940 |
| PMMA | 2,450 | 2,450 | 1,200 |
| POM | 5,000 | 5,000 | 1,400 |
| Balsa | 50 | 50 | 150 |

**Table 9.** Materials used for the structural components of the W- and S-model blades.

| Component | From (% span) | To (% span) | Material type | |
|---|---|---|---|---|
| | | | W-model | S-model |
| External shell | 0 | 100 | Bx GFRP | PMMA |
| Spar caps | 10 | 95 | Ux GFRP | POM |
| Shear web | 10 | 95 | Bx GFRP | PMMA |
| TE/LE reinforcements | 10 | 45 | Ux GFRP | PMMA |
| Shell and web core | 10 | 95 | Balsa | Balsa |

For the S-model, the thermoplastic materials polymethyl methacrylate (PMMA) and polyoxymethylene (POM) are chosen
because of their lower stiffness. The use of polymer materials reduces the nonstructural masses, as the adhesive is no longer necessary. Due to the reduced fatigue characteristics of these materials, the blade lifetime is limited to 5 years. This is assumed to be acceptable in the present case, given the research nature of these blades. Constraints on maximum stresses and strains are satisfied with ample margin for these blades. However, the inclusion of a larger set of DLCs (including extreme events and parked conditions) might create more challenging situations, which could increase the requirements on material strength,
possibly eventually leading to the selection of different materials.

Figure 4 reports the internal structure of the W- and S-models, as well as the overall mass distributions, including realistic nonstructural masses. The scaled mass distribution follows quite closely the reference one along the blade span, with the exception of the root because of the additional thickness that must be ensured to accommodate the bolted connection. The blade satisfies the scaling inertial and elastic constraints within a tolerance of less than 5%.

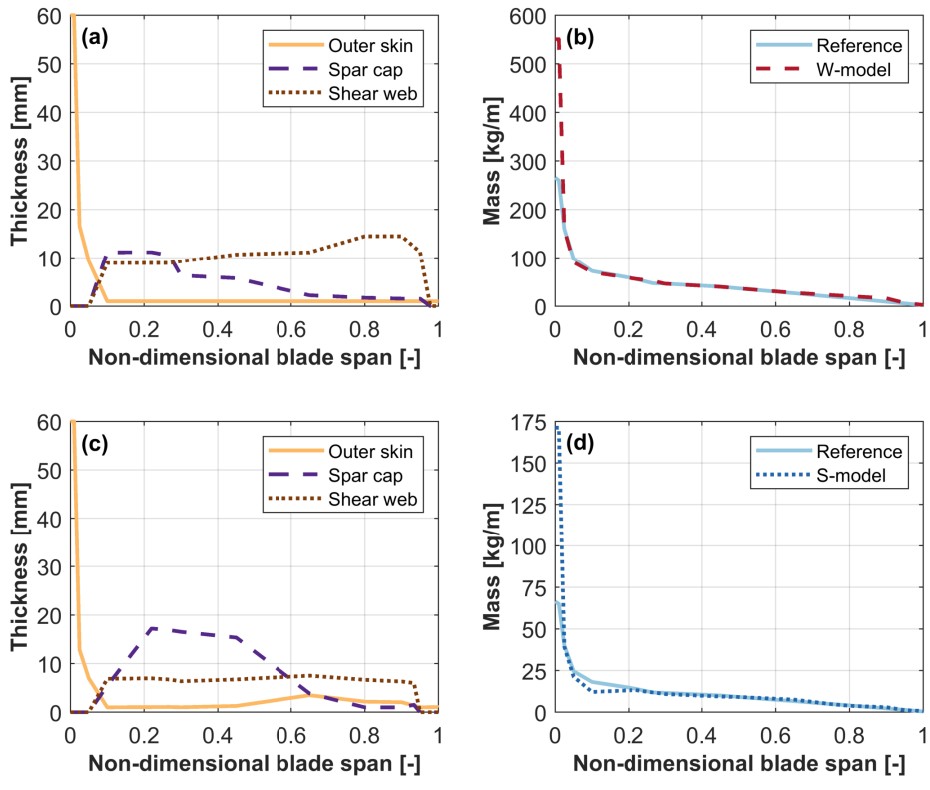

**Figure 4.** Thickness of the structural components and mass distribution for the W- (top) and S- (bottom) models. The label "reference" indicates the mass distribution of the full-scale blade, subscaled to the W- and S- scales.

### 4.3.3 Redesign of the T-model

The very small size of the wind tunnel model blade prevents the use of a typical thin-walled solution. Following Bottasso et al. (2014) and Campagnolo et al. (2014), this scaled blade is not hollow, but presents a full cross section obtained by machining a foamy material. Two unidirectional spar caps provide the required flapwise stiffness distribution. The surface smoothness is obtained by a very thin layer of skin made of glue. Although Bottasso et al. (2014) and Campagnolo et al. (2014) considered different scaling laws, their blade design configuration was found to be a suitable choice even in the present gravo-aeroservoelastic scaling exercise. The selection of appropriate materials represents a critical aspect of the problem, and the mechanical properties listed in the Cambridge University Materials Data Book (Materials Data Book, 2003) were used to guide the material selection for the spar caps and core. A rigid polymer foam is chosen as filler, because of its relatively high stiffness and lightness. For the spar caps, thermoplastic polymers are again found to be the most suitable solution even though their stiffness-to-density ratio is much lower than materials traditionally used for spar caps. Moreover, the use of thermoplastics allows for alternative

and simpler manufacturing processes, leading to a higher flexibility in the spar cap design. From this family of materials, polypropilene is chosen because of its low stiffness modulus. Finally, the external shell is covered by a very thin layer of the epoxy structural adhesive Scotch Weld AF 32 (3M, 2000).

The design variables are represented by the spanwise thickness and width of the two spars. The design problem is formulated according to the constrained matching optimization expressed by Eq. (18). The cost function of Eq. (19) considers the spanwise mass distribution in $\mathcal{S}_M$ and the flapwise stiffness distribution in $\mathcal{S}_K$. The matching constraints $\mathbf{m}_s$ include the lowest three natural frequencies, and the flapwise static extreme tip deflection. Both the cost and the constraints only consider the flapwise characteristics of the blade, because the structural configuration consisting of a solid core and two spar caps allows for a limited control of the edgewise characteristics. As a result, the scaled blade presents a higher edgewise stiffness than the full-scale reference.

Figure 5 reports the results of the design optimization. The desired matching of mass and flapwise stiffness is achieved, except at blade root. Even though the placement of the first flapwise natural frequency with respect to the rotor speed is ensured, the constraint on the lowest edgewise natural frequency could not be exactly matched due to the large chord. Small disparities in mass distribution introduce a difference of about 1% in the blade flapping inertia.

## 5 Performance comparison

In this section, the behavior of the scaled models is compared to the full-scale machine. The main goal here is to assess to which extent the subscale models are capable of successfully mirroring relevant key characteristics and load trends of the full-scale reference.

The same collective-pitch/torque controller governs all machines. The controller uses a look-up table for torque to operate at rated TSR in region II, and a proportional-integral-derivative (PID) pitch loop to maintain constant rated power in region III. The PID gains used for the scaled models are obtained by transforming the ones of the full-scale machine using the scaling laws, and the regulation trajectory is adapted to each model to account for differences in the $C_P$-TSR curves. Notice that the scaling of gains is a conservative approach: in the case of an exact matching at scale of all aeroelastic characteristics of the turbines, the use of a scaled controller will ensure also identical closed-loop response. However, if the scaled models do not represent exactly the full-scale reference —which is invariably the case in practice— an ad hoc retuned controller (i.e., a controller specifically optimized for the scaled model) will in general have a better performance than the one obtained by the scaling of the gains. The choice of gain scaling instead of retuning was made here to consider a worst-case scenario.

### 5.1 Relevant key indicators

The models are simulated in a power production state at five different wind speeds from cut-in to cut-out. The winds of the scaled simulations are obtained by velocity scaling the turbulent winds used for the full-scale machine (i.e. the integral space and timescales are both correctly scaled). The matching between the scaled and full-scale turbines is assessed with the help of 10 different indicators: annual energy production (AEP), maximum flapwise tip displacement (MFTD), maximum thrust

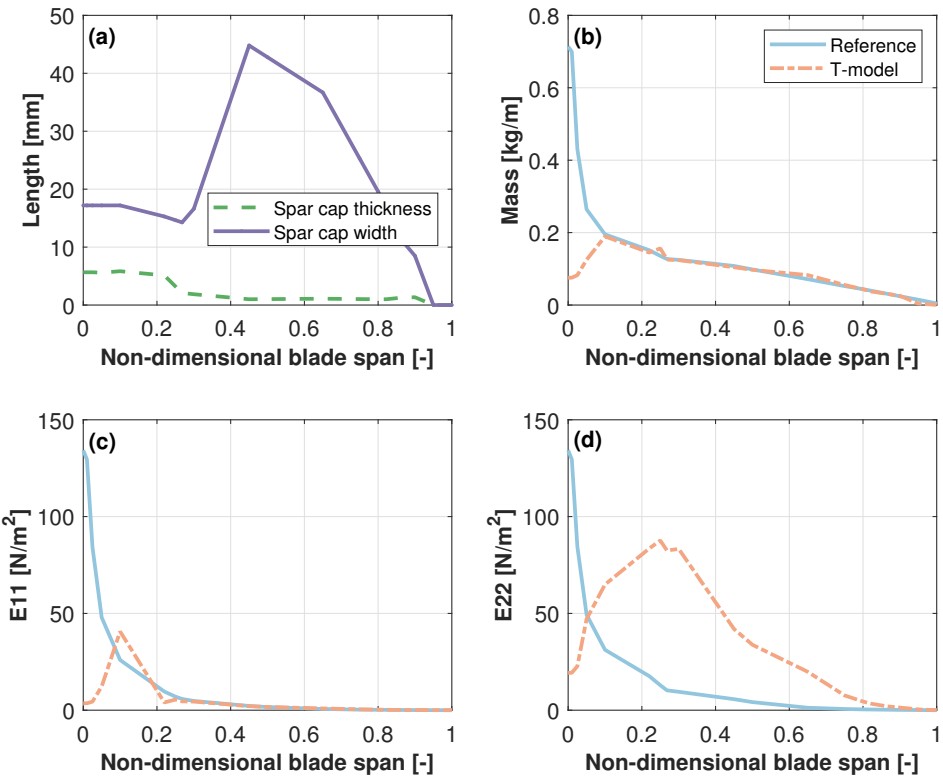

**Figure 5.** (a) Spar caps thickness and width, (b) mass distribution, (c) flapwise stiffness distribution, and (d) edgewise stiffness distribution for the T-model. The label "reference" indicates the characteristics of the full-scale blade, subscaled to the T-model scale.

at main shaft (ThS), maximum combined blade root moment (CBRM), maximum flapwise bending root moment (FBRM), maximum edgewise bending root moment (EBRM), and the Weibull-averaged damage equivalent loads (DEL) for ThS, CBRM, FBRM and EBRM.

### 5.1.1 Utility-scale models

As previously discussed, both the design of the external shape and of the internal structure may induce differences in the behavior of a scaled model with respect to its full-scale reference. To better understand the effects of these differences and their origins, three different sets of results are presented in Fig. 6.

The first plot (at top left) compares the indicators of the full-scale turbine with the upscaled ones of the W- and S-models. Both the internal structure and the external shape are obtained by zooming, and Reynolds effects are accounted for by CFD-computed polars. Although a zoomed-down structure cannot really be a practical solution —as discussed earlier— because of excessively thin structural elements or the need for peculiar material properties, this solution is shown here because it

highlights the sole effects of the Reynolds mismatch. In other words, since this is a purely numerical study, the thicknesses
and mechanical properties were used exactly as produced by scaling, resulting in a nearly exact satisfaction of the matching
of all structural characteristics. Therefore, the differences of the indicators between the full-scale and scaled models shown in
this plot can be entirely attributed to Reynolds effects. The full-scale and utility-size models are equipped with airfoil polars at
different Reynolds computed with the CFD code `ANSYS Fluent` (ANSYS, Inc., 2019).

The second plot (at top right) compares the indicators for the W- and S-models with a zoomed-down external shape, but
neglecting Reynolds effects, and a redesigned internal structure. Although Reynolds effects would, in reality, be present, by
neglecting them here —which is again possible because this is a purely numerical study— one can assess from this solution
the sole effects of the structural redesign on the matching of the indicators.

Finally, the third and last plot (bottom part of the figure) considers the solution obtained by zooming down the aerodynamic
shape, considering Reynolds effects, and a redesigned internal structure. As argued earlier, this is indeed the solution that is
720 practically realizable, and, therefore, these are the more realistic results of the set considered here. Hence, differences between
the full-scale and scaled models are due to mismatches caused both by Reynolds and the redesign procedure.

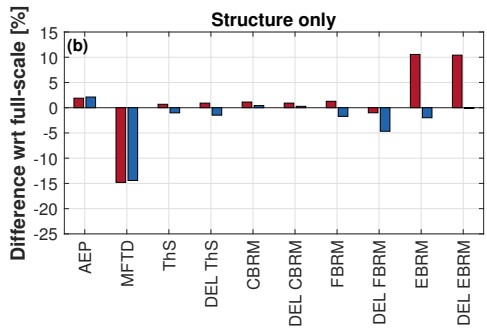

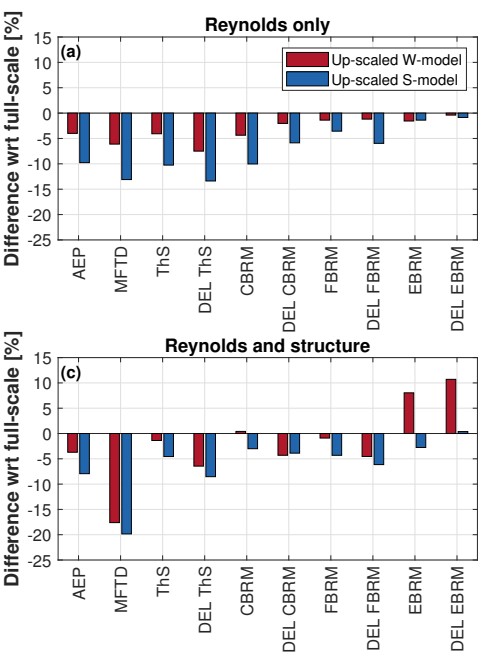

**Figure 6.** Changes with respect to full scale for several key indicators for the W- and S-models. (a) Effects exclusively due to Reynolds mismatch, (b) effects exclusively due to structural redesign, and (c) realistic solution considering both the effects of Reynolds mismatch and structural redesign.

As expected by the size difference, results shown in the first plot suggest a larger effect of the Reynolds mismatch for the
S-model than for the W-model. This results in a drop in all indicators because of the decreased airfoil efficiency.

The second plot shows a similar matching for both models. Indeed, most of the key loads are matched within 5% for both the W- and the S-model. A larger difference between the two models is found for EBRM and DEL EBRM, which are only poorly matched by the W-model, whereas they are quite accurate for the S-model. The mismatch is due to a slightly higher sectional mass in the last 20% of the blade of the W-model, as shown in Fig. 4. A significant difference with respect to full scale is also observed for the maximum flapwise tip displacement of both the W- and S-models. This difference is caused by a slightly different dynamic behavior induced by mismatches in the flapwise and torsional stiffness distributions. Even though FBRM matches very well for both the W- and S-model at the root, these differences lead to a poorer match at sections toward the blade tip, which in the end impacts MFTD.

Overall, both models are capable of matching the key indicators of the full-scale target reasonably well, considering both Reynolds effects and a redesigned structure.

### 5.1.2 Wind tunnel model

The behavior of the T-model is compared with the 10 MW baseline in Fig. 7. The additional indicator maximum edgewise tip deflection (METD) is considered in this case. The polars for the T-model are computed with Xfoil (Drela, 2013).

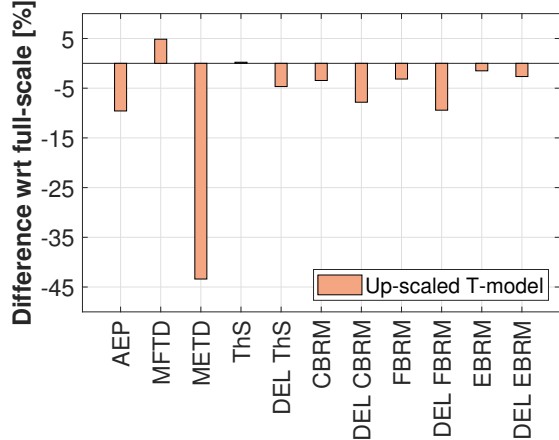

**Figure 7.** Comparison between full-scale key indicators and the upscaled ones of the T-model.

The comparison shows satisfactory behavior of the wind tunnel model for most key indicators, notwithstanding the very different Reynolds numbers (about 1E+7 for the full-scale reference, and about 2E+4 for the T-model). As expected, the largest mismatch is found for the maximum edgewise tip displacement. This can be justified by the inability of the structural design variables (limited to the two caps) in controlling the edgewise stiffness.

## 5.2 Load trends in waked conditions

Scaled models can also be used to capture trends, instead of absolute values. Indeed, the goal of scaled testing is often to understand the trends generated on some metric by, for example, a control technology, or by a particular operating condition or other factors, whereas the exact quantitative assessment of the induced effects must be left to a final full-scale verification.

As an example of the analysis of trends, the scaled models designed here are used to explore changes in loading between unwaked and waked inflow conditions. To this end, the full-scale turbine is simulated with an average inflow velocity of 7 ms$^{-1}$, considering a shear exponent of 0.2 and a turbulence intensity of 8%. The wake deficit generated by an upstream 10 MW machine is then added to this inflow, in order to simulate a waked condition. The wake is modeled by the superposition of a turbulent wind grid generated with `TurbSim` (Jonkman et al., 2009) and the first order solution of the deficit of the Larsen model (EWTSII model) (Bottasso et al., 2017). The downstream turbine is located at a longitudinal downstream distance of 4 D from the upstream machine, and its lateral distance from the wake center is varied from -1.25 D (right, looking downwind) to 1.25 D (left), realizing different degrees of wake-rotor overlap. The scaled models are simulated by velocity-scaling the full-scale inflows. The key indicators considered are AEP, ThS, FBRM and DEL for CBRM, FBRM and EBRM.

Figure 8 reports changes in key indicators at several degrees of wake overlap with respect to unwaked inflow conditions. The full-scale machine presents the largest reduction in AEP and ThS in full wake overlap. An asymmetrical load trend of the DELs for FBRM, EBRM and CBRM is visible when the rotor is operating in partial wake. This behavior is mostly due to the rotor uptilt angle, which introduces an additional velocity component in the rotor plane. In fact, for a clockwise (when looking downstream) rotating rotor, this extra velocity component increases the in-plane velocity at the blade sections when the blade is on the right side of the rotor (i.e., during the downstroke; here left and right are defined for an observer looking downstream). Additionally, when a wake impinges on the right side of the rotor, the out-of-plane velocity component decreases, because of the wake deficit. Both of these effects tend to decrease the angle of attack at the blade sections. On the other hand, when a wake impinges on the left portion of the rotor, the effect of the decreased out-of-plane component is in part balanced by the also decreased in-plane component. Because of this different behavior, larger load fluctuations (and hence higher fatigue loads) are observed for right wake impingements than for left ones. A similar effect is caused by the elasticity of the tower: under the push of the thrust, the tower bends backwards that in turn tilts the rotor upward, adding to the previously described phenomenon. Other minor effects are also due to the elastic deformations caused by gravity, which again contribute to breaking the symmetry of the problem.

Overall, the largest scaled models follow the trends very well, with the S-model performing slightly better than the W-model. Indeed, the W-model is better than the S-model when looking at Weibull-averaged quantities (Fig. 6), but the S-model presents a slightly superior matching of blade loads at the specific speed at which the load trend study is performed. The trends are also reasonably captured by the smaller-scale T-model, but with significant differences in DEL FBRM. Specifically, there is an overestimation of this quantity around the $-0.5$ D lateral wake center position. A detailed analysis of the results revealed this behavior to be caused by the blade operating at angles of attack close to the stalling point. This indicates another possible

limit of models with large-scale factors, whose airfoils may have very different stall and post-stall behavior than their full-scale

counterparts.

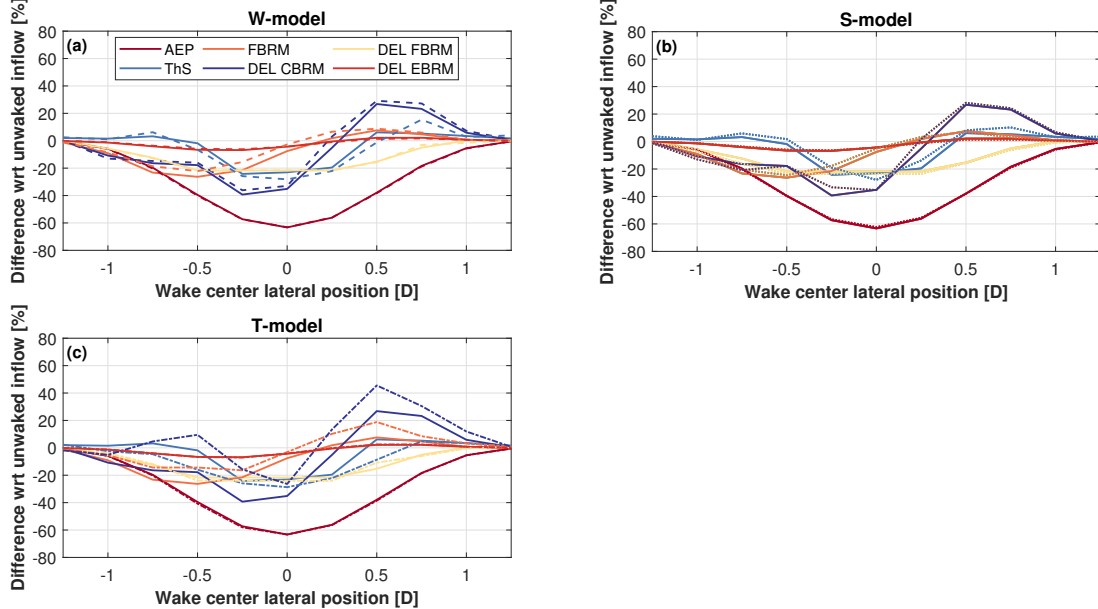

**Figure 8.** Comparison of key indicators between unwaked and waked inflows, for different lateral distances from the wake center. The solid line corresponds to the full-scale model. (a) W-model (dashed line); (b) S-model (dotted line); (c) T-model (dash-dotted line).

## 6  Conclusions

This paper analyzed the scaling conditions that should be met by a subscale model to match a full-scale reference in terms of its full aeroservoelastic response. The analysis has shown that many relevant key aspects of the steady and unsteady response of a machine, considered as flexible, can indeed be matched. Part of this analysis can also be used to understand expected

changes due to upscaling, which can be useful in the design of larger rotors. To the authors' knowledge, this is one of the most comprehensive analysis of the problems of scaling wind turbines presented thus far.

Within this framework, this paper has considered two alternative ways of designing a scaled rotor. The first is based on the idea of exactly zooming down the full-scale reference to obtain the subscale model. An alternative strategy is to completely redesign the rotor, both from an aerodynamic and structural point of view. This produces a scaled blade that, although possibly

very different from the full-scale one, matches some of its key characteristics as closely as possible.

These two alternative strategies have been tested on the gravo-aeroservoelastic scaling of a conceptual 10 MW blade to three different subscale models: two utility-scale ones to be used for the reblading of small existing turbines, and one for equipping a very small model turbine for conducting experiments in the controlled environment of a wind tunnel.

The following conclusions can be drawn from the application of the two strategies to these three different scaling problems.

The simplest strategy to design the external shape of utility-scale blades is the straightforward zooming-down approach, as long as the subscale Reynolds is sufficiently high. This strategy benefits from a simple implementation and leads to an acceptable match of the blade aerodynamic performance. However, when the blade aerodynamic performance is compromised by the Reynolds mismatch —which is the typical case of wind tunnel models— the alternative but more complex strategy of redesigning the aerodynamic shape becomes preferable if not altogether indispensable. Special low-Reynolds airfoils may be used to mitigate the effects caused by the reduced Reynolds regime. However, different behavior at and around stall might lead to different loads when operating at large angles of attack.

The straightforward zooming down of the blade internal structure is instead typically very difficult for all scaling ratios. In fact, the need for materials of unusual characteristics and the nonscalability of nonstructural masses unfortunately hinder the applicability of this simple approach. An alternative is found in the structural redesign strategy, which offers more flexibility at the price of increased complexity. Even here, however, the problem is nontrivial. For example, materials may play a critical role, due to the very flexible nature of some of these scaled blades.

The aeroservoelastic analyses conducted herein have shown that, in general, it is not possible to exactly match all the characteristics of a full-scale machine with a subscale model. However, with the proper choices, some key indicators are nicely captured. In addition, changes in operating conditions are represented quite well even at the smaller scale. For example, it was shown that changes in loading from an unwaked to a waked condition are accurately represented by all scaled models, which successfully capture intricate and possibly unexpected couplings with design aspects such as nacelle uptilt and tower deflection. The good performance of the models in capturing such complex effects opens up a range of applications and use cases. For example, with the right design choices, scaled models can be employed to better understand rotor-wake interactions or test sophisticated control strategies at the turbine and/or plant levels.

Further improvements in the performance of the subscale models are certainly possible. Indeed, while some of the limitations result from the choice of quantities to be matched, others can be overcome by technological advances. For instance, improvements in measurement technology can relax the requirements on the scaling of time, allowing for a better match of other quantities. Additionally, advances in material and manufacturing may ease the application of unconventional materials, relax sizing constraints, and lead to more accurate, simpler, faster to develop and cheaper models.

This work has exclusively focused on the wind turbine itself, and the effects of scaling have been quantified for the aerodynamic performance and loading of the rotor. The recent study of Wang et al. (2020) expands this analysis by considering the effects of scaling on wake behavior. Even in that case the conclusion is that properly scaled models can produce very realistic wakes.

Further work should focus on expanding the scope of the scaling analysis, introducing the effect of hydrodynamics. Indeed, as floating wind energy is expected to significantly grow in the coming years, it becomes increasingly important to better understand which aspects of the aero-hydroservoelastic response of these machines can be matched, and how to best design subscale models. This is however only part of the problem. Research efforts are also necessary to better understand how to replicate the inflow conditions that full-scale machines face in various types of atmospheric and terrain conditions. This is a

challenging task, since it requires a deep understanding of atmospheric flows, their interaction with the terrain orography and the vegetation, and technology to replicate these flows at scale.

It is the hope of the authors that the results shown in this paper will increase the confidence on scaled testing, in the belief that scaled model have a significant role to play in the advancement of wind energy science.

*Code and data availability.* The data used for the present analysis can be obtained by contacting the authors.

*Author contributions.* HC modified the `Cp-Max` code to support the scaled matching optimization, designed the subscale models, performed the simulations and analyzed the results; CLB devised the original idea of this research, performed the theoretical scaling analysis, formulated the matching optimization problem and supervised the work; PB collaborated in the modification of the software, the design of the subscale models and the conduction of the numerical simulations. HC and CLB wrote the manuscript. All authors provided important input to this research work through discussions, feedback and by improving the manuscript.

*Competing interests.* The authors declare that they have no conflict of interest.

*Acknowledgements.* The authors would like to thank Chengyu Wang and Daniel J. Barreiro of the Technical University of Munich for the computation of the airfoil polars using CFD for multiple Reynolds numbers. Additionally, credit goes to Eric Loth of the University of Virginia for having introduced the authors to the zooming approach, and to Filippo Campagnolo of the Technical University of Munich for the fruitful discussions and support. This work was authored in part by the National Renewable Energy Laboratory, operated by Alliance for Sustainable Energy, LLC, for the U.S. Department of Energy (DOE) under Contract No. DE-AC36-08GO28308. Funding provided by the U.S. Department of Energy Office of Energy Efficiency and Renewable Energy Wind Energy Technologies Office. The views expressed in the article do not necessarily represent the views of the DOE or the U.S. Government. The U.S. Government retains and the publisher, by accepting the article for publication, acknowledges that the U.S. Government retains a nonexclusive, paid-up, irrevocable, worldwide license to publish or reproduce the published form of this work, or allow others to do so, for U.S. Government purposes.

*Financial support.* This work is funded in part by the WINSENT project (FKZ: 0324129F), which receives funding from the German Federal Ministry for Economic Affairs and Energy (BMWi).

**Nomenclature**

| | |
|---|---|
| $a$ | Axial induction factor |
| $a_s$ | Speed of sound |

| | | |
|---|---|---|
| | $c$ | Chord length |
| 850 | $d$ | Out-of-plane blade section flapping displacement |
| | $f$ | Characteristic frequency |
| | $g$ | Acceleration of gravity |
| | $l$ | Characteristic length |
| | $n_l$ | Geometric scaling factor, i.e. $l_M/l_P$ |
| 855 | $n_t$ | Time scaling factor, i.e. $t_M/t_P$ |
| | $n_\Omega$ | Angular velocity scaling factor, i.e. $\Omega_M/\Omega_P$ |
| | $n_v$ | Wind speed scaling factor, i.e. $V_M/V_P$ |
| | $\mathbf{p}$ | Vector of design parameters |
| | $r$ | Spanwise coordinate |
| 860 | $s$ | Tip deflection |
| | $t$ | Time |
| | $u$ | Characteristic speed |
| | $A$ | Rotor disk area |
| | $A_b$ | Blade planform area |
| 865 | $B$ | Number of blades |
| | $C_D$ | Drag coefficient |
| | $C_L$ | Lift coefficient |
| | $C_{L,\alpha}$ | Slope of the lift curve |
| | $C_P$ | Power coefficient |
| 870 | $C_T$ | Thrust coefficient |
| | $E$ | Young's modulus or airfoil efficiency $C_L/C_D$ |
| | $EJ$ | Bending stiffness |
| | $Fr$ | Froude number |
| | $I$ | Rotor polar moment of inertia |
| 875 | $I_b$ | Blade flapping inertia |
| | $J$ | Cost function |
| | $K$ | Stiffness |
| | $Lo$ | Lock number |
| | $M$ | Mass |
| 880 | $Ma$ | Mach number |
| | $P$ | Aerodynamic power |
| | $Q$ | Torque |
| | $R$ | Rotor radius |
| | $Re$ | Reynolds number |
| 885 | $Ro$ | Rossby number |
| | $St$ | Strouhal number |

| | | |
|---|---|---|
| | $T$ | Thrust force |
| | $U_P$ | Flow velocity component perpendicular to the rotor disk plane |
| | $U_T$ | Flow velocity tangent to the rotor disk plane |
| 890 | $V$ | Wind speed |
| | $W$ | Flow speed relative to a blade section |
| | $\beta$ | Blade pitch |
| | $\epsilon$ | Strain |
| | $\theta$ | Sectional pitch angle |
| 895 | $\kappa$ | Reduced frequency |
| | $\lambda$ | Tip-speed ratio |
| | $\lambda_d$ | Design TSR |
| | $\mu$ | Fluid dynamic viscosity |
| | $\nu$ | Poisson coefficient |
| 900 | $\rho$ | Air density |
| | $\rho_m$ | Material density |
| | $\rho_P$ | Power density |
| | $\sigma$ | Stress |
| | $\tau$ | Nondimensional time |
| 905 | $\omega$ | Natural frequency |
| | $\Gamma$ | Circulation |
| | $\Delta\theta$ | Total blade twist from root to tip |
| | $\Sigma$ | Rotor solidity |
| | $\Phi$ | Rotor uptilt angle |
| 910 | $\Xi$ | Rotor cone angle |
| | $\Omega$ | Rotor angular velocity |
| | $(\cdot)_a$ | Pertaining to the aerodynamic design |
| | $(\cdot)_s$ | Pertaining to the structural design |
| | $(\cdot)_M$ | Scaled system |
| 915 | $(\cdot)_P$ | Full-scale physical system |
| | $\dot{(\cdot)}$ | Derivative with respect to time, i.e. $\mathrm{d}\cdot/\mathrm{d}t$ |
| | $(\cdot)'$ | Derivative with respect to nondimensional time, i.e. $\mathrm{d}\cdot/\mathrm{d}\tau$ |
| | $\widetilde{(\cdot)}$ | Nondimensional quantity |
| | $\widehat{(\cdot)}$ | To-be-matched scaled quantity |
| 920 | AEP | Annual energy production |
| | BEM | Blade element momentum |
| | Bx | Biaxial |
| | CBRM | Combined bending root moment |
| | CFD | Computational fluid dynamics |

| 925 | CFRP | Carbon-fiber-reinforced plastic |
| | DEL | Damage equivalent load |
| | DLC | Design load case |
| | EBRM | Edgewise bending root moment |
| | FBRM | Flapwise bending root moment |
| 930 | GFRP | Glass-fiber-reinforced plastic |
| | LD | Low density |
| | LE | Leading edge |
| | MFTD | Maximum flapwise tip displacement |
| | METD | Maximum edgewise tip displacement |
| 935 | PID | Proportional integral derivative |
| | PMMA | Polymethil methacrylate |
| | POM | Polyoxymethylene |
| | PP | Polypropilene |
| | SQP | Sequential quadratic programming |
| 940 | ThS | Thrust at main shaft |
| | TSR | Tip-speed ratio |
| | TE | Trailing edge |
| | Tx | Triaxial |
| | Ux | Uniaxial |

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
