# Peer review of "On the scaling of wind turbine rotors"

_Wind Energy Science, 2020_

## Referee Comment (RC1) · Anonymous Referee #1 · 11 May 2020

**General comments**
It is a very well written paper and has good contribution to the scientific progress of wind energy. The authors have presented a very comprehensive methodologies, discussions and results on the aero-elastic scaling of the wind turbine rotor. It answered clearly the 3 common scientific questions mentioned in the very beginning of this paper regarding the scaling of a wind turbine rotor. The reviewer still has some specific comments to address in order to further improve the quality of the paper.

**Specific comments**
In the following, there are some more technical comments:

1. On page 1, line 24, "an alternative design approach". What exactly is the design approach?

2. On page 7, line 9, " Hence, non-dimensional deflections can always be matched, provided that the stiffness is adjusted as shown". But the stiffness can not always be adjusted as it needs easily due to the limitation on the material properties. The author should consider to strengthen this argument.

3. On page 11, line 5 "If the model is actuated (with generator, pitch and yaw), it becomes increasingly difficult if not altogether impossible to house these systems in the reduced dimensions of the model." It is difficult to understand this sentence. What does the author mean?

4. On page 16, line 1 "For instance, the standard blades of the V27 weigh 600 kg (Vestas, 1994); four times more than the gravo-aeroservoelastically scaled blades of the S-model." The author should consider or mention that the V27 blade was designed 15 years ago using relatively old technology, which should be heavier than a blade designed by newer technology.

5. On page 17, line 9 " as efficiency is still relatively high" What is your reference case for this statement?

6. On page 17, line 11 "the FFA-W3-241 airfoil behaves very poorly." Could you please show a figure here?

7. On page 17, line 12 "because its aerodynamic characteristics at the scaled Reynolds are in reasonable agreement with the ones of the original airfoil at its full-scale Reynolds." Could you show a figure to support your argument?

8. On page 19, line 3 "The third web of the full-scale blade is also extremely thin (less than 1 mm) and very close to the trailing edge." This sentence is miss-leading. If I understand correctly, should it be the blade of W-model or S-model?

9. On page 19, line 7 "For example, the outer shell requires an elasticity modulus of 6.6 GPa and a density of 1,845 " Is this statement made for which sub-scaled

blade? W-model or S-model?

10. On page 19, line 31 "matrices". What matrices? Could you please be more detailed?

11. On page 20, Figure 3. Too much information is provided in this figure. If you could remove some of the non-relevant info, you could improve the clarity of the figure.

12. On page 20, line 13-14. Why extreme loads are not considered?

13. On page 23, Figure 5. Could you explain more in detailed about the "reference" used in figure 5?

14. On page 24, line 7-8 "The proportional-integral-derivative gains used for the scaled models are obtained by scaling the ones of the full-scale machine," Why and how do you scale these PID gains? In my opinion, a good method is to re-tune them. Could you explain on why to you scale them instead of re-tune?

15. On page 24, line 22 "up-scaled". From my understanding, should this be down-scaled?

16. On page 24, section 5, The wake model used for calculate wake deficit is not mentioned. Could you briefly describe it?

17. On page 26, line 3-4, "The mismatch is due to a slightly higher sectional mass in the last 20

18. On page 26. What about the comparison of the natural frequencies? Could you please show one plot regarding the frequencies in this section?

19. On page 27, line 7-8, Which wake model is used?

20. On page 27, line 13, The critical explanation of the results shown in figure 8 is missing.

21. In general, The results and conclusions reflect the outcome of this research work well. But some statement is missing, for example, it was not mentioned how the rated wind speed, rotor speed were selected during the sub-scaling design process?

**Technical corrections**
Some of the small grammar mistake and typos are found and listed from my side:

1. On page 3, line 34, "aeroelastically" -> aero-elastically

2. On page 19, line 14, on -> in

3. On page 19, line 29, composites -> composite; appear -> appears

4. On page 27, line 18, overestimation -> over estimation

---

## Referee Comment (RC2) · Anonymous Referee #2 · 31 Aug 2020

General Comments:

This paper is quite well written and covers a wide spectrum of scaling effects to be considered when conducting experiments on scaled turbines. A particular strength of the work is in examining the scaling of a very large rotor to several sizes, from near-utility to wind-tunnel. This reviewer has a few specific and technical comments to improve the paper.

Specific Comments:

Page 10, line 2: While stiffness can be changed to some extent through material substitutions and laminate sizing, and non-structural mass can be added, the effects are seen in both flap and edge directions, so some trade-off will likely need to be made depending on the scaled phenomena and modes in question. The author could elaborate on this issue as it would likely be critical to scaling aero-elastic instabilities, for

instance.

Page 16, Table 7: For scaled turbines, the tower height and stiffness is fixed. This influences the dynamics of the rotor system. How does the scaling of the rotor take this into account?

Page 21, line 10: Fatigue is mentioned here, but without data to evaluate the claim. Also, material strength is not discussed as a limitation. This seems unlikely to be true. The operational loads of the turbine can be modified to some extent by the controller, but there are still parked loads that are quite high. This would be a very practical issue with a scaled rotor in a field environment.

Technical Comments: Page 18, line 8: Perhaps "issues", "difficulties" or "challenges" is a better word choice than "aspects"

---

## Author Comment (AC1) · 18 Nov 2020

**REVISION TO MANUSCRIPT DRAFT**

**Wind Energy Science Discussion**

**On the scaling of wind turbine rotors**

The authors would like to thank the two reviewers for their time and for the useful feedback. All their inputs have been taken into consideration, and have contributed to the improvement of the paper. In addition, we have taken the opportunity of this revision to make several editorial changes in order to improve readability, and we have expanded the text at various points throughout the manuscript to improve clarity.

A revised version of the paper is attached to the present reply, with the main changes highlighted in red (deletions) and blue (additions).

A list of point-by-point replies to the reviewers' comments is reported in the following.

**Reviewer #1**

**Numbered comments**

1. *[Reviewer]* *On page 1, line 24, "an alternative design approach". What exactly is the design approach?*

   **[Authors]** We believe there is a typo and the reviewer means page 3, line 24. The alternative design approach refers to the complete aerostructural redesign of the blade external shape and internal structure. The sentence has been modified to improve clarity.

2. *[Reviewer]* *On page 7, line 9, "Hence, non-dimensional deflections can always be matched, provided that the stiffness is adjusted as shown". But the stiffness can not always be adjusted as it needs easily due to the limitation on the material properties. The author should consider to strengthen this argument.*
   **[Authors]** We agree, and this is one of the main challenges in the design of scaled models. To solve the problem, very often scaled models are designed with a different structural configuration and choice of materials than the original full-scale system. To clarify this point, three new references have been added (Wan and Cesnik, 2014; Ricciardi et al., 2016; Busan, 1998), and section 3.2 has been significantly expanded, adding a detailed mathematical formulation of the re-design problem. The description of the redesigned models has also been expanded (correcting also some imprecisions that were present in the previous version). These modifications also address comment #1 of Reviewer #2.

3. *[Reviewer]* *On page 11, line 5 "If the model is actuated (with generator, pitch and yaw), it becomes increasingly difficult if not altogether impossible to house these systems in the reduced dimensions of the model." It is difficult to understand this sentence. What does the author mean?*
   **[Authors]** The paragraph has been rewritten to improve clarity.

4. *[Reviewer]* *On page 16, line 1 "For instance, the standard blades of the V27 weigh 600 kg (Vestas, 1994); four times more than the gravo-aeroservoelastically scaled blades of the S-model." The author should consider or mention that the V27 blade was designed 15 years ago using relatively old technology, which should be heavier than a blade designed by newer technology.*
   **[Authors]** Thank you for this remark. The text has been modified to address this point.

5. *[Reviewer] On page 17, line 9 "as efficiency is still relatively high" What is your reference case for this statement?*

    **[Authors]** The authors mean that the FFA-W3-241 airfoil at the typical Reynolds numbers of the W- and S-models is still performing reasonably well with respect to the airfoil performance at full-scale. We agree that this sentence can be misleading, and we have reformulated it to improve clarity.

6. *[Reviewer] On page 17, line 11 "the FFA-W3-241 airfoil behaves very poorly." Could you please show a figure here?*
    **[Authors]** This sentence has been replaced by a more general explanation of the need to adopt a low-Reynolds airfoil. We believe it is no longer necessary to show a figure of the performance of the FFA-W3-241 airfoil at the Reynolds number of the T-model.

7. *[Reviewer] On page 17, line 12 "because its aerodynamic characteristics at the scaled Reynolds are in reasonable agreement with the ones of the original airfoil at its full-scale Reynolds." Could you show a figure to support your argument?*

    **[Authors]** The efficiency and polars of the RG14 airfoil at the typical Reynolds number of the T-model have been added to Figure 1.

8. *[Reviewer] On page 19, line 3 "The third web of the full-scale blade is also extremely thin (less than 1 mm) and very close to the trailing edge." This sentence is misleading. If I understand correctly, should it be the blade of W-model or S-model?*
    **[Authors]** Thank you for spotting this error. The sentence refers to the structure of the blade of the S-model. The text has been corrected.

9. *[Reviewer] On page 19, line 7 "For example, the outer shell requires an elasticity modulus of 6.6 GPa and a density of 1,845 " Is this statement made for which sub-scaled blade? W-model or S-model?*
    **[Authors]** The statement refers to the W-model. The text has been improved for clarity.

10. *[Reviewer] On page 19, line 31 "matrices". What matrices? Could you please be more detailed?*
    **[Authors]** Thank you for spotting this typo. The sentence has been corrected.

11. *[Reviewer] On page 20, Figure 3. Too much information is provided in this figure. If you could remove some of the non-relevant info, you could improve the clarity of the figure.*
    **[Authors]** Thank you for this suggestion, we completely agree. We have replaced the plot by a simpler one to improve clarity.

12. *[Reviewer] On page 20, line 13-14. Why extreme loads are not considered?*

    **[Authors]** Maximum stresses and strains are computed from extreme loads. The scaled models presented in this work were designed only considering extreme loads from DLC 1.1 (power production with normal turbulence model), but a more detailed design should be based on extreme loads resulting from a larger set of DLCs, including operating conditions in extreme events and stand still conditions. The paragraph has been expanded to address this point.

13. *[Reviewer] On page 23, Figure 5. Could you explain more in detailed about the "reference" used in figure 5?*
    **[Authors]** The lines marked "reference" in Figures 4 and 5 display characteristics (e.g. stiffness and mass distributions) of the full-scale blade, sub-scaled with the corresponding scaling factors. This clarification has now been added to the caption of Figures 4 and 5.

**14.** *[Reviewer] On page 24, line 7-8 "The proportional-integral-derivative gains used for the scaled models are obtained by scaling the ones of the full-scale machine," Why and how do you scale these PID gains? In my opinion, a good method is to re-tune them. Could you explain on why you scale them instead of re-tune?*
**[Authors]** Both scaling and re-tuning of the controllers are possible methods to define the control strategy of the scaled models. We chose the first option because it is the conservative one, whereas an ad-hoc re-optimization of the controllers might hide some mismatched characteristic of the scaled turbines. We have expanded the text to better explain this point.

**15.** *[Reviewer] On page 24, line 22 "up-scaled". From my understanding, should this be downscaled?*
**[Authors]** The various computed quantities of all sub-scale models are up-scaled to the full scale. They are then directly compared to the corresponding quantities of the full-scale model.

**16.** *[Reviewer] On page 24, section 5, The wake model used for calculate wake deficit is not mentioned. Could you briefly describe it?*
**[Authors]** The wake is modeled by the superposition of a turbulent wind grid generated with TurbSim and the first order solution of the wind speed deficit of the Larsen model (EWTSII model). This explanation has now been added to the text.

**17.** *[Reviewer] On page 26, line 3-4, "The mismatch is due to a slightly higher sectional mass in the last 20 [sic]*

**[Authors]** This sentence refers to the slightly higher sectional mass of the W-model around the section positioned at 90% of blade span, which can be observed in Fig. 4. However, the sentence seems truncated and we do not understand the reviewer's question.

**18.** *[Reviewer] On page 26. What about the comparison of the natural frequencies? Could you please show one plot regarding the frequencies in this section?*

**[Authors]** The first three nondimensional natural frequencies of the W- and S-models are placed with a tolerance of 5% respect the full-scale ones. For the T-model, the placement of the edgewise frequency is well above the reference value due to the very large chord. The placement of the natural frequencies for all models is given in the text.

A figure displaying the comparison of natural frequencies for the three models is shown here:

[Figure]

However, we have not included this figure in the manuscript, since the same information can be found in the text, and the paper is already quite long and with many figures.

**19.** *[Reviewer] On page 27, line 7-8, Which wake model is used?*

**[Authors]** See comment #16.

20. **[Reviewer]** *On page 27, line 13, The critical explanation of the results shown in figure 8 is missing.*
**[Authors]** Thank you for pointing this out. We have now added a critical explanation of the results shown in Figure 8.

21. **[Reviewer]** *In general, The results and conclusions reflect the outcome of this research work well. But some statement is missing, for example, it was not mentioned how the rated wind speed, rotor speed were selected during the sub-scaling design process?*

**[Authors]** The rated wind speed and rated rotor speed are defined with the scaling laws. Indeed, the sub-scale models are designed to have the same TSR as the full-scale machine. The rated rotor speed can then be automatically derived as:

$$\Omega_M = \frac{\Omega_P}{n_t}$$

If the Cp of the sub-scale model is equal to the full-scale model one, the rated speed would scale as follows:

$$V_{rated\,M} = V_{rated\,P}\frac{n}{n_t}$$

The rated wind speed is adapted for each model to account for differences in Cp-TSR curves. We have included this information to section 4.2.

**Technical corrections**

1. **[Reviewer]** On page 3, line 34, "aeroelastically" -> aero-elastically

    **[Authors]** We believe "aeroelastically" is consistent with the spelling guidelines used in the article.

2. **[Reviewer]** On page 19, line 14, on -> in
    **[Authors]** Thank you, the typo has been corrected.

3. **[Reviewer]** On page 19, line 29, composites -> composite; appear -> appears

    **[Authors]** The sentence refers to both biaxial and uniaxial glass-fiber-reinforced plastic composites, therefore we believe it is correct to use the plural form.

4. **[Reviewer]** On page 27, line 18, overestimation -> over estimation

    **[Authors]** We believe "overestimation" is consistent with the spelling guidelines used in the article.

**Reviewer #2**

**Specific comments**

1. **[Reviewer]** *Page 10, line 2: While stiffness can be changed to some extent through material substitutions and laminate sizing, and non-structural mass can be added, the effects are seen in both flap and edge directions, so some trade-off will likely need to be made depending on the scaled phenomena and modes in question. The author could elaborate on this issue as it would likely be critical to scaling aero-elastic instabilities, for* instance*.*

    **[Authors]** We agree that the stiffness adjustment might not always be straightforward and it might be necessary to make some compromises to overcome the challenges it presents (such as

the choice of suitable materials). We have expanded the text to better explain the matching design problem, as already noted in replying to comment #2 of Reviewer #1.

2. **[Reviewer]** *Page 16, Table 7: For scaled turbines, the tower height and stiffness is fixed. This influences the dynamics of the rotor system. How does the scaling of the rotor take this into account?*
**[Authors]** The tower characteristics listed in Table 7 correspond to the full-scale tower characteristics sub-scaled with the corresponding scaling factors (listed in Table 6). This paper focuses on scaling approaches for the design of the wind turbine rotor and assumes the external and internal characteristics of other components such as tower, drivetrain or actuators to be perfectly scaled according to the scaling factors. However, we agree that it might not always be possible to adopt components whose dimensions and characteristics perfectly follow the scaling laws. The adoption of functional larger components might affect the turbine behavior. If this affects quantities that should be accurately represented at scale, different scaling factors should be chosen. We modified part of section 2.3 to address this comment, as well as comment #3 of Reviewer #1.

3. **[Reviewer]** *Page 21, line 10: Fatigue is mentioned here, but without data to evaluate the claim. Also, material strength is not discussed as a limitation. This seems unlikely to be true. The operational loads of the turbine can be modified to some extent by the controller, but there are still parked loads that are quite high. This would be a very practical issue with a scaled rotor in a field environment.*
**[Authors]** This is a very good point, thank you for bringing it up. The design of the W- and S-blades is based on extreme loads resulting from DLC 1.1 and material strength was not identified as a limitation. However, the inclusion of a larger set of DLCs (including extreme events and parked conditions) will create more challenging situations that might increase the requirements. In this case, requirements on material strength should be considered during material selection. This point is now discussed in section 4.3.2.

**Technical comments:**

1. **[Reviewer]** *Page 18, line 8: Perhaps "issues", "difficulties" or "challenges" is a better word choice than "aspects"*
*"Aspects" has been replaced by "challenges"*

The authors

[revised manuscript text omitted]

---

## Editor Decision (ED1)

2020-66

Overall
- This is a really nice paper on a relevant and interesting topic and a good overall contribution. However, the overall framing is a bit weak. I would consider rewriting the abstract and thinking a bit more about what the key contributions of this paper are compared to the state of the art. They are stronger than the abstract indicates.
- Also, generally there is a lot of detail and in depth discussion but the paper lacks higher level synthesis which would be helpful at several key points in the paper (discussed in more detail below). Help the reader orient on the key takeaways throughout.

Abstract
- Why is the study framed as a comparison of geometric versus scaling to match key performance indicators? The issues associated with geometric scaling are widely understood. It does not seem a particularly strong way to frame the work

Introduction
- The list of questions seems ahead of itself. It would be better to start with the motivation for why we do scaling, how has it been done before and what are the challenges with doing so.
- Reference on the machine scaling past 200 m? references on reasons for upscaling? Reference on the square cubed law? There is lots of stuff out there so it would be good to reference something
- Subscaling motivations incomplete – there are a few key reasons for this:
    o Cost – with the size of turbines today, one cannot simply build a new concept for every interesting innovation idea
    o Control in testing – field testing is challenging because you cannot control the conditions and get the range of types of test conditions you might want. Wind tunnels offer the ability to replicate a broad range of conditions but you of course have to scale down the turbine model even for the largest wind tunnels in the world
- The discussion on the limitations of matching characteristics when subscaling to wind tunnel scale seems incomplete – what can and can not be matched? What trade-offs have to be made? And why can some of those be captured at the scale of the SWIFT field testing facility? – its addressed later so maybe mention that it will be
- V27 blade size and rated power worth mentioning since it is not self evident for those not familiar

Scaling
- Very nice overall discussion, before jumping into the subsections, could be nice to give a bit of a high-level overview (before the start of section 2.1)
- End of section 2 could also be stronger. The reader does not get a clear sense of the challenges and trade-offs in scaling. The anecdote on the nacelle is good but it would be stronger to use something tied to the blade design (for example – using the swift blades, or others)

Design strategies
- Why is the aerodynamic design problem focused on Cp error minimization as the objective function? Any of the performance metrics could be the objective and Cp matching could be a constraint… worth explaining choice
- This section could benefit from a brief additional subsection comparing and contrasting the geometric and aerodynamic design strategies on a theoretical basis.

10 MW subscaling
- Similarly here, a summary would be helpful – there is a lot of detailed information and there is a bit of a challenge seeing the forest for the trees

Performance comparison
- Consider a different title for section 5.2 – i.e. Load trends under waked conditions
- Section 5.2 is SUPER interesting and deserves a bit more discussion and prominence in the overall article… consider expanding on it and using as a key part of the overall framing and motivation for the paper

Conclusions
- Missing a decent recommendation on future work – for example, 5.2 looked at trends in loads from effects of upstream wakes, what about the control strategies applied on both upstream and downstream turbines?
- Also, there could be a bit more of circling back on the ties to issues of scaling with of atmospheric structures. Even if we can solve the turbine scaling issues, that is only part of the problem

---

## Author Response (AR2)

**Reply to Editor**

Wind Energy Science Discussion

Paper: "On the scaling of wind turbine rotors"

We provide here a list of point-by-point replies to the comments of the Editor.

**Overall**

1. **[Editor]** *This is a really nice paper on a relevant and interesting topic and a good overall contribution. However, the overall framing is a bit weak. I would consider rewriting the abstract and thinking a bit more about what the key contributions of this paper are compared to the state of the art. They are stronger than the abstract indicates. Also, generally there is a lot of detail and in depth discussion but the paper lacks higher level synthesis which would be helpful at several key points in the paper (discussed in more detail below). Help the reader orient on the key takeaways throughout.*
   **[Authors]** We have partly modified the abstract and the introduction to address these comments.

**Abstract**

2. **[Editor]** *Why is the study framed as a comparison of geometric versus scaling to match key performance indicators? The issues associated with geometric scaling are widely understood. It does not seem a particularly strong way to frame the work*
   **[Authors]** Clearly the presentation could be done in a different way. Yet, in our opinion the comparison between the apparently simple zooming down and the more complex redesign is very interesting and makes for a much more informative presentation. In fact, simple zooming the external aerodynamic shape works well up until Reynolds effects force towards different solutions, while structural zooming is much more constrained. For example, we are not sure that the effects of changes of planform shape and solidity are well known and widely understood (nor we are aware of similar in depth discussions available in the literature), especially because they have intricate effects of rotational augmentation, optimum TSR, vorticity shedding etc., as explained in Sect. 2 and then later in the paper. In our opinion contrasting these two methods is more enriching, informative and convincing for the reader, than simply adopting a redesign approach without giving a motivation for it. In fact, several interesting comparisons and comments are made all throughout Sect. 5, which would be largely lost by dropping the zooming approach from the presentation.
   In any case, we have reworded and expanded parts of the introduction in order to address these comments.

**Introduction**

3. **[Editor]** *The list of questions seems ahead of itself. It would be better to start with the motivation for why we do scaling, how has it been done before and what are the challenges with doing so.*
   **[Authors]** Even though starting the paper with the list of research questions might not follow the conventional style, we believe it is the most beneficial way for the reader in this case. Due to the significant length of the paper, we believe it is useful for the reader to get a clear picture of the goals and scope of paper before diving into the motivation and background of the work.

4. **[Editor]** *Reference on the machine scaling past 200 m? references on reasons for upscaling? Reference on the square cubed law? There is lots of stuff out there so it would be good to reference something*
**[Authors]** We have added references and expanded the discussion, thanks for the suggestion.

5. **[Editor]** *Subscaling motivations incomplete – there are a few key reasons for this:*
   - *Cost – with the size of turbines today, one cannot simply build a new concept for every interesting innovation idea*
   - *Control in testing – field testing is challenging because you cannot control the conditions and get the range of types of test conditions you might want. Wind tunnels offer the ability to replicate a broad range of conditions but you of course have to scale down the turbine model even for the largest wind tunnels in the world*

   **[Authors]** We totally agree, we simply tried to keep this part not too long, as we have written extensively on these topics in the past. However, we have now extended the discussion to give more emphasis to these motivations in the introduction.

6. **[Editor]** *The discussion on the limitations of matching characteristics when subscaling to wind tunnel scale seems incomplete – what can and can not be matched? What trade-offs have to be made? And why can some of those be captured at the scale of the SWIFT field testing facility? – its addressed later so maybe mention that it will be*
   **[Authors]** We agree with this comment, however we believe it is premature to discuss the limitations of subscaling to wind tunnel size in the introduction, which is already very long, also because it needs technical information that has not yet been presented. We have added a comment indicating that this topic will be discussed later in the article.

7. **[Editor]** *V27 blade size and rated power worth mentioning since it is not self evident for those not familiar*
   **[Authors]** Thank you for the suggestion, we completely agree and have added the rated power and blade size.

**Scaling**
8. **[Editor]** *Very nice overall discussion, before jumping into the subsections, could be nice to give a bit of a high-level overview (before the start of section 2.1)*
   **[Authors]** Thank you for the suggestion, we have added an overview before the start of section 2.1.

9. **[Editor]** *End of section 2 could also be stronger. The reader does not get a clear sense of the challenges and trade-offs in scaling. The anecdote on the nacelle is good but it would be stronger to use something tied to the blade design (for example – using the swift blades, or others)*
   **[Authors]** We added a paragraph to emphasize the strong limitations in the choice of scaling parameters. We also added a comment about how certain combinations of geometric and time scaling factor might lead to design requirements that can be challenging to fulfill.

**Design strategies**
10. **[Editor]** *Why is the aerodynamic design problem focused on Cp error minimization as the objective function? Any of the performance metrics could be the objective and Cp matching could be a constraint… worth explaining choice. This section could benefit from a brief additional subsection*

*comparing and contrasting the geometric and aerodynamic design strategies on a theoretical basis.*

**[Authors]** The formulation of the aerodynamic design problem depends on the specific application of the sub-scale model, so different formulations are certainly possible. Although this was already expressed in the text, we have now reinforced this concept and rephrased a few sentences for improved clarity.

**10 MW subscaling**

**11.** [Editor] *Similarly here, a summary would be helpful – there is a lot of detailed information and there is a bit of a challenge seeing the forest for the trees*
**[Authors]** We have added a paragraph introducing the following sections.

**Performance comparison**

**12. [Editor]** *Consider a different title for section 5.2 – i.e. Load trends under waked conditions*
**[Authors]** Thank you, the change has been implemented.

**13. [Editor]** *Section 5.2 is SUPER interesting and deserves a bit more discussion and prominence in the overall article… consider expanding on it and using as a key part of the overall framing and motivation for the paper*
**[Authors]** The goal of the section is to assess the performance of the sub-scale models under conditions with complex dynamics, such as the intricate wake-rotor interaction shown here. We believe the section currently fulfills its purpose, and it is therefore not necessary to expand it. However, we agree that the results are very interesting and should be further highlighted. We have added a few sentences in the abstract, introduction and conclusion to emphasize the good performance of the scaled models even under these complex conditions.

**Conclusions**

**14. [Editor]** *Missing a decent recommendation on future work – for example, 5.2 looked at trends in loads from effects of upstream wakes, what about the control strategies applied on both upstream and downstream turbines?*
**[Authors]** We have expanded the conclusion section and included a few recommendations for future work.

**15. [Editor]** *Also, there could be a bit more of circling back on the ties to issues of scaling with of atmospheric structures. Even if we can solve the turbine scaling issues, that is only part of the problem*
**[Authors]** Thank you for this suggestion, we have added a few lines addressing this need.

We have taken the opportunity to make several small editorial changes to the text, in order to improve readability. A revised version of the manuscript is attached to the present reply, with the main additions highlighted in blue and deletions in red.

Best regards,
The authors

[revised manuscript text omitted]